# A*-Thought: Efficient Reasoning via Bidirectional Compression for Low-Resource Settings

Xiaoang Xu[1]  Shuo Wang[2*]  Xu Han[2,4,5]  Zhenghao Liu[3]
Huijia Wu[1]  Peipei Li[1]  Zhiyuan Liu[2,4,5]  Maosong Sun[2,4,5]  Zhaofeng He[1*]
[1]Beijing University of Posts and Telecommunications
[2]Dept. of Comp. Sci. & Tech., Tsinghua University, Beijing, China
[3]Northeastern University  [4]Institute for AI, Tsinghua University, Beijing, China
[5]Beijing National Research Center for Information Science and Technology

## Abstract

Large Reasoning Models (LRMs) achieve superior performance by extending the thought length. However, a lengthy thinking trajectory leads to reduced efficiency. Most of the existing methods are stuck in the assumption of overthinking and attempt to reason efficiently by compressing the Chain-of-Thought, but this often leads to performance degradation. To address this problem, we introduce A*-Thought, an efficient tree search-based unified framework designed to identify and isolate the most essential thoughts from the extensive reasoning chains produced by these models. It formulates the reasoning process of LRMs as a search tree, where each node represents a reasoning span in the giant reasoning space. By combining the A* search algorithm with a cost function specific to the reasoning path, it can efficiently compress the chain of thought and determine a reasoning path with high information density and low cost. In addition, we also propose a bidirectional importance estimation mechanism, which further refines this search process and enhances its efficiency beyond uniform sampling. Extensive experiments on several advanced math tasks show that A*-Thought effectively balances performance and efficiency over a huge search space. Specifically, A*-Thought can improve the performance of QwQ-32B by $2.39\times$ with low-budget and reduce the length of the output token by nearly 50% with high-budget. The proposed method is also compatible with several other LRMs, demonstrating its generalization capability. The code can be accessed at: `https://github.com/AI9Stars/AStar-Thought`.

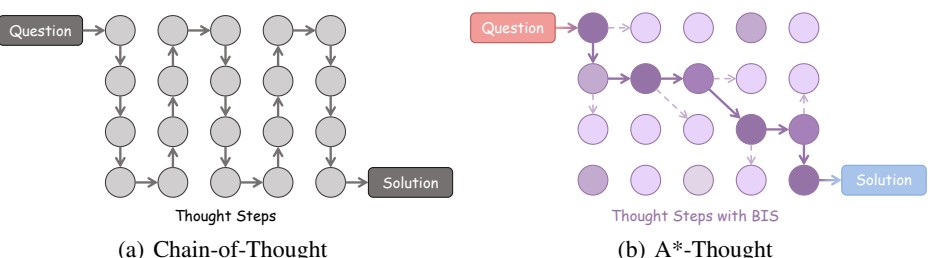

(a) Chain-of-Thought          (b) A*-Thought

Figure 1: Illustration of the comparison between the standard CoT and the proposed A*-Thought. In A*-Thought, each thinking step is assigned a bidirectional importance score (BIS), represented by varying color shades. Guided by the carefully-designed cost functions, A*-Thought efficiently arrives at the solution using fewer steps, reducing the redundancy inherent in the original CoT.

---

* Corresponding authors.

39th Conference on Neural Information Processing Systems (NeurIPS 2025).

# 1  Introduction

Large Reasoning Models (LRMs), such as o1 (OpenAI et al., 2024), R1 (DeepSeek-AI, 2025) and QwQ (Qwen Team, 2025), have demonstrated remarkable advancements in performance on a variety of complex tasks. These significant leaps are largely attributable to their capacity for sophisticated long-form reasoning, often operationalized through the generation of extended chain-of-thought (CoT) sequences. However, this enhanced reasoning capability introduces a substantial trade-off in terms of computational overhead during inference (Xia et al., 2025; Aggarwal & Welleck, 2025). The generation and processing of these lengthy CoTs inherently demand greater storage and computational resources. Consequently, the escalating costs associated with these long CoTs pose a considerable barrier to the practical application and widespread adoption of LRMs, particularly in resource-constrained environments such as on-device or end-point deployments (Hu et al., 2024; Allal et al., 2025) where computational budgets and memory footprints are strictly limited.

To address these efficiency concerns, a growing body of recent research has begun to investigate long-to-short methodologies. These approaches aim to significantly condense the reasoning processes inherent in LRMs, thereby enhancing their inference efficiency (Lee et al., 2025b; Han et al., 2025; Aytes et al., 2025; Xia et al., 2025; Qu et al., 2025; Hao et al., 2024). For example, Chen et al. (2025) identify and conceptualize the overthinking phenomenon in LRMs, where models may engage in unnecessarily verbose or circuitous reasoning. They propose to mitigate this by specifically training the models to favor more concise responses. In a complementary effort, Xiang et al. (2025) introduce a prompting strategy that guides LRMs to articulate their reasoning in shorter, more atomic steps, which cumulatively leads to a reduction in the total length of the generated thought process.

Rather than addressing a singular, narrowly defined issue within LRMs, this work introduces a unified framework designed to identify and isolate the most essential thoughts from the extensive reasoning chains produced by these models. An effective intermediate thinking step should ideally possess two core properties: (1) *High-quality*: The step's logical progression and substantive content must be accurate, sound, and directly relevant to the specific context and boundaries of the question. (2) *Informativeness*: The step must make a demonstrable contribution towards the target solution by furnishing critical information. This characteristic holds even if an intermediate conclusion, while perhaps imperfect or erroneous in isolation, ultimately facilitates the derivation of the correct final answer. The main challenges are two-fold:

> 1. How to identify high-quality and informative steps at the *step-level*?
> 2. How to effectively assemble individual reasoning steps into a concise and effective reasoning path at the *path-level*?

We propose A*-Thought, a novel framework for automatically discovering compact and effective reasoning paths by leveraging signals at both step and path levels. At the step level, a bidirectional importance estimation mechanism quantifies the significance of each thinking step based on its relevance to both the question and the prospective solution. At the path level, A* search is employed to efficiently navigate the exponential search space. This search utilizes cost functions that assess two key aspects: the quality of the current path and the conditional self-information of the solution given this path. These assessments collectively inform the estimated current and future cost to reach a desirable final solution. Experimental results demonstrate that A*-Thought successfully learns effective LRMs across diverse inference budgets, surpassing several representative baselines.

The primary contributions of this research are delineated as follows:

- We design a step-level bidirectional importance score (BIS) to evaluate the criticality of individual sentences. This scoring mechanism serves to significantly enhance effectiveness of the A* search procedure compared to standard sampling techniques.
- We introduce a path-level A* search algorithm tailored for compressing lengthy CoTs from LRMs. This algorithm strategically considers both current path quality and estimated future costs to optimize LRM performance under stringent output length constraints.
- The proposed algorithm demonstrates substantial empirical gains over several representative baselines. For instance, for QwQ-32B in concise output scenarios, it achieves a $3.53\times$ improvement in accuracy.

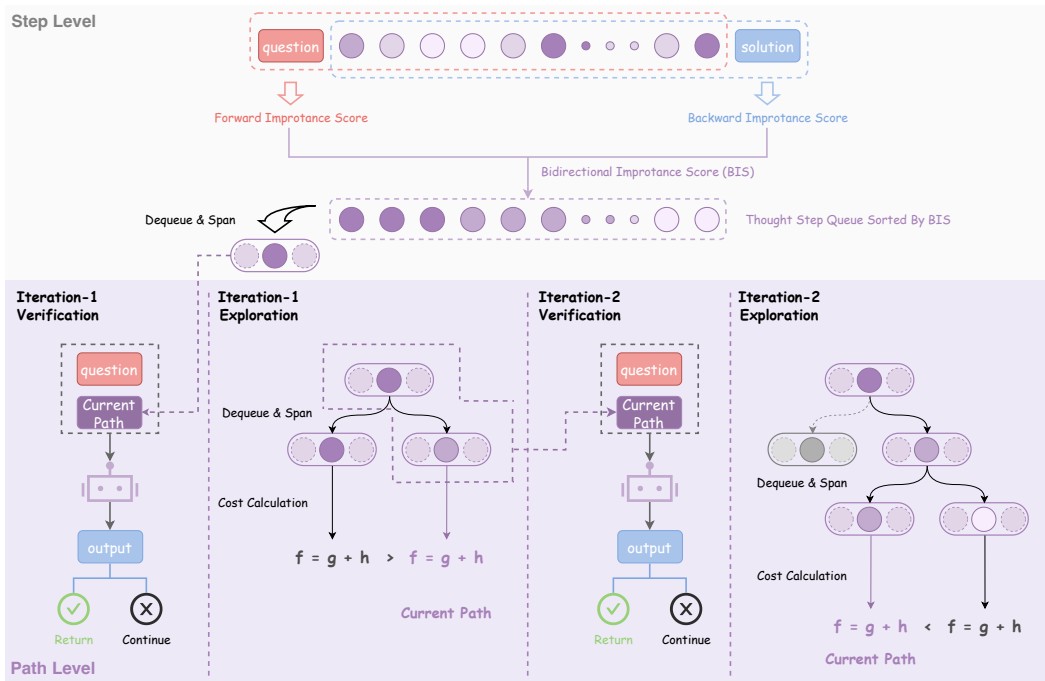

Figure 2: Illustration of A*-Thought, a long-CoT compression method. A*-Thought leverages signals at both the step and path levels. At the step-level, a bidirectional importance score assesses relevance to both the question and the solution. At the path-level, an A* search algorithm is employed, with cost functions designed to consider both current path quality and estimated future cost.

## 2 Problem Setup

Given an LRM $\mathcal{M}$, it can generate an extended thinking trajectory comprising $N$ reasoning steps, denoted as $\mathbf{t} = \left\{ \mathbf{t}^{(1)}, \mathbf{t}^{(2)}, \ldots, \mathbf{t}^{(N)} \right\}$, and the corresponding solution $\mathbf{s}$ for a given question $\mathbf{q}$. This process can be represented as:

$$(\mathbf{t}, \mathbf{s}) := \mathcal{M}(\mathbf{q}). \tag{1}$$

However, as the number of reasoning steps $N$ increases, the model may frequently switch its thinking modes. Sampling a complete thinking trajectory can hinder convergence of the solution estimation, thereby increasing the inference cost.

Our objective is to identify a subset $\mathbf{t}' \subseteq \mathbf{t}$ that preserves the LRM's reasoning performance to the greatest extent possible. A more compact thinking trajectory $\mathbf{t}'$ is expected to exhibit a higher knowledge density across various computational budgets.

## 3 Method

In this section, we first introduce the bidirectional importance score of each local thinking step, which is employed in the subsequent A* search process for thought compression.

### 3.1 Step-Level Bidirectional Importance Score

As mentioned in Section 2, it is prohibitively costly to explore all potential subsets of the long CoT. We therefore propose an importance score to improve sampling efficiency for thought selection. A related approach is LongLLMLingua (Jiang et al., 2024), which employs conditional probabilities to estimate the question-aware importance of each token in a long document. In complex tasks that require long CoTs, the importance of an intermediate thought is not only related to the question but also to the final solution. Relevance to the solution may provide important information for identifying useful thoughts. We therefore propose a bidirectional importance score that considers both the question and the final solution when estimating the importance of each thinking step.

Specifically, we assess importance at two levels, including the attention and model levels. At the attention level, we use attention weights to represent the importance of $\mathbf{x}$ for $\mathbf{y}$:

$$\text{ATTN}(\mathbf{y}|\mathbf{x}) = \frac{1}{H|\mathbf{y}||\mathbf{x}|} \sum_{h=1}^{H} \sum_{j=1}^{|\mathbf{y}|} \sum_{i=1}^{|\mathbf{x}|} a_h(y_j, x_i), \quad (2)$$

where $a_h(y_j, x_i)$ denotes the attention score of the query $y_j$ to the key $x_i$ for the $h$-th head. $H$ represents the number of heads. A higher importance score indicates a more significant effect of $\mathbf{x}$ on $\mathbf{y}$. At the model level, we use the negative log-likelihood to assess the importance score of $\mathbf{x}$ for $\mathbf{y}$:

$$\text{NLL}(\mathbf{y}|\mathbf{x}) = -\frac{1}{|\mathbf{y}|} \sum_{j=1}^{|\mathbf{y}|} \log P(y_j|\mathbf{x}, \mathbf{y}_{<j}). \quad (3)$$

To assess the contribution of each thinking step $\mathbf{t}^{(n)}$ in the overall thought t, we analyze its influence on both the question $\mathbf{q}$ and the solution $\mathbf{s}$. Each thought is concatenated with the question and the solution, forming the sequences $\langle \mathbf{t}^{(n)}, \mathbf{q} \rangle$ and $\langle \mathbf{t}^{(n)}, \mathbf{s} \rangle$, respectively. Subsequently, we utilize a compact language model, specifically GPT-2[2], to quantify attention scores and NLL values. The selection of a smaller model enhances the efficiency of this importance estimation procedure.

Specifically, the bidirectional importance score of $\mathbf{t}^{(n)}$ is denoted as

$$\text{BIS}\left(\mathbf{t}^{(n)}\right) = \frac{(1-\alpha)\,\text{ATTN}\left(\mathbf{q}|\mathbf{t}^{(n)}\right) + \alpha\text{ATTN}\left(\mathbf{s}|\mathbf{t}^{(n)}\right)}{(1-\alpha)\,\text{NLL}\left(\mathbf{q}|\mathbf{t}^{(n)}\right) + \alpha\text{NLL}\left(\mathbf{s}|\mathbf{t}^{(n)}\right)}, \quad (4)$$

where $\alpha$ is a hyper-parameter to control the relative weighting of relevance to the question versus the solution. Figure 3 presents the distribution of BIS for a sequence of thought steps, where the thought steps are divided by "\n\n". The figure illustrates that not all steps hold high importance concerning both the question and the solution. Only a limited subset offers significant contributions. Subsequently, these BIS values will be leveraged in conjunction with the A* search algorithm to determine the pruned thought $\mathbf{t}'$.

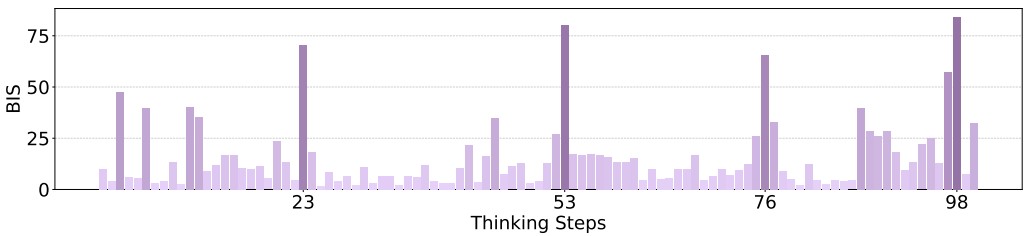

Figure 3: Distribution of BIS values for individual thinking steps in Long CoT.

## 3.2 Path-Level A* Search

Our objective is to extract an alternative thought trajectory $\mathbf{t}'$ from an original trajectory $\mathbf{t}$. For a trajectory $\mathbf{t}$ with $N$ thinking steps, the $2^N$ candidate trajectories render exhaustive exploration computationally intractable for extended $\mathbf{t}$. We therefore employ the A* search algorithm for efficient traversal of this space. Following an initialization phase, the algorithm iteratively conducts verification and exploration. In the $k$-th iteration, a verification model $\mathcal{V}$ ascertains if the current candidate path $\mathbf{t}'_k$ can lead to the correct solution. During exploration, each path $\mathbf{t}'_k$ is evaluated using cost functions. The algorithm is detailed in the following subsections.

### 3.2.1 Overview

To enhance search efficiency, the thinking steps within the original long CoT $\mathbf{t} = \left\{\mathbf{t}^{(1)}, \mathbf{t}^{(2)}, \ldots, \mathbf{t}^{(N)}\right\}$ are first sorted in descending order based on their BIS values, yielding

---
[2]`https://huggingface.co/openai-community/gpt2`.

$\mathbf{t}_{\text{sort}} = \left\{ \mathbf{t}^{(n_1)}, \mathbf{t}^{(n_2)}, \dots, \mathbf{t}^{(n_N)} \right\}$. Subsequently, the A* search algorithm iteratively expands a search tree, denoted as $\mathcal{T}$, according to defined cost functions. In this tree, each node corresponds to a span centered on a specific thinking step, encompassing its immediately preceding and succeeding steps. Formally, a node associated with the thinking step $\mathbf{t}^{(n)}$ is represented as $\mathbf{r}^{(n)} = \langle \mathbf{t}^{(n-1)}, \mathbf{t}^{(n)}, \mathbf{t}^{(n+1)} \rangle$. This approach aims to mitigate the adverse effects of fragmented information that can arise from thought segmentation.

**Initialization**  By leveraging the bidirectional importance estimation mechanism detailed in Section 3.1, A*-Thought identifies a logical starting point within the potentially redundant thinking trajectory, thereby enhancing both efficiency and performance. Initially, a thought queue, denoted as $\mathcal{Q}$, is constructed using the sorted thought sequence $\mathbf{t}_{\text{sort}}$. Subsequently, the first thought is dequeued from $\mathcal{Q}$ to form the root node of the search tree $\mathcal{T}$. This selection ensures the implementation of a best-first sampling strategy throughout the subsequent search iterations.

**Verification**  To assess the efficacy of the current thinking path, denoted as $\mathbf{t}'_k$, which encompasses the thinking spans from the root node to the current active leaf node within the search tree $\mathcal{T}$, a validation model $\mathcal{V}$ is introduced. This model is employed to determine whether the current path $\mathbf{t}'_k$ successfully leads to the solution $\mathbf{s}$:

$$\mathcal{V} \left( \mathbf{q} + \mathbf{t}'_k \right) \begin{cases} \neq \mathbf{s}, & \text{expand } \mathcal{T} \\ = \mathbf{s}, & \text{return } \mathbf{t}' = \mathbf{t}'_k \end{cases} \tag{5}$$

It has been observed that verification tends to be ineffective for extremely short thought sequences, thereby offering limited guidance for the search process. Consequently, a lower boundary, denoted as $k_{\min}$, is established for verification. Verification is performed exclusively when the depth of the search tree, $k$, satisfies the condition $k \geq k_{\min}$.

**Exploration**  If the current active leaf node does not pass verification, the first $W$ thoughts are dequeued from $\mathcal{Q}$ to function as next-level leaf nodes, denoted as $\{\mathbf{r}_1, \dots, \mathbf{r}_W\}$. Each of these nodes is then appended to the current thinking path $\mathbf{t}'_k$ to construct a set of candidate thinking paths. we assign a cost function $f(\cdot)$ to each candidate thinking path, where

$$f\left(\mathbf{t}'_k + \mathbf{r}_w\right) = g\left(\mathbf{t}'_k + \mathbf{r}_w\right) + h\left(\mathbf{t}'_k + \mathbf{r}_w\right). \tag{6}$$

The design of our cost function, $f(\cdot)$, is informed by the A* search algorithm. Specifically, $g(\cdot)$ denotes the cumulative cost incurred from the root node to the current node. Concurrently, $h(\cdot)$ functions as a heuristic, providing an estimate of the prospective cost from the current node to the target solution. We select the node that with the minimal cost as the new active leaf node:

$$\hat{\mathbf{r}}_w = \underset{w \in \{1, \cdots, W\}}{\arg\min} f\left(\mathbf{t}'_k + \mathbf{r}_w\right). \tag{7}$$

The newly formed active thinking path, $\mathbf{t}'_{k+1} = \langle \mathbf{t}'_k, \hat{\mathbf{r}}_w \rangle$, subsequently proceeds to the next iteration of the process. Figure 2 shows an example for the search process. To prevent an excessively deep search tree, an upper bound, $k_{\max}$, is imposed on its depth. The search process is terminated, and $\mathbf{t}$ is directly returned when the current depth, $k$, reaches or exceeds this limit, i.e., $k \geq k_{\max}$. The resulting compact responses can be leveraged to distill LRMs, fostering enhanced thinking efficiency.

### 3.2.2  Design of Cost Functions

To identify an effective and compact thought, denoted as $\mathbf{t}'$, the quality of each intermediate thought $\mathbf{t}'_k$ is assessed from two perspectives: (1) the quality of the current intermediate thought, which is quantified by $g(\cdot)$; and (2) the estimated future cost associated with extending the current intermediate thought $\mathbf{t}'_k$ to the final thought sequence $\mathbf{t}'$.

**Current Cost Function**  The function $g(\cdot)$ measures the quality of the current intermediate thought $\mathbf{t}'_k$. A verification model leveraging its reasoning capabilities is employed to estimate this quality:

$$g\left(\mathbf{t}'_k\right) = -\frac{\beta}{|\mathbf{t}'_k|} \log P_{\mathcal{V}}\left(\mathbf{t}'_k | \mathbf{q}\right), \tag{8}$$

where $\beta$ is the weight controlling the effect of the current cost.

**Future Cost Function** The function $h(n)$ estimates the cost from the current node to the goal, thereby influencing the efficiency of the search path. A higher estimated future cost suggests that a more extensive sequence of future thoughts will be required to reach the final solution from the current state. To quantify this, we employ the conditional self-information of the correct solution $\mathbf{s}$, given the current thought $\mathbf{t}'_k$ and the input question $\mathbf{q}$. Formally, it can be represented as:

$$h(\mathbf{t}'_k) = \mathcal{I}\left(\mathbf{s}|\mathbf{q}, \mathbf{t}'_k\right). \tag{9}$$

Larger values of the conditional self-information $\mathcal{I}(\cdot)$ indicate a lower likelihood of generating the solution $\mathbf{s}$. We quantify $\mathcal{I}(\cdot)$ as follows:

$$\mathcal{I}\left(\mathbf{s}|\mathbf{q}, \mathbf{t}'_k\right) = -\frac{1}{|\mathbf{s}|} \log P_{\mathcal{V}}\left(\mathbf{s}|\mathbf{q}, \mathbf{t}'_k\right). \tag{10}$$

In particular, A*-Thought enhances reasoning efficiency by compressing the thought trajectory. This is achieved through the systematic reduction of redundant steps, thereby streamlining the path from the initial query to the final solution. Such targeted compression significantly improves the performance of LRMs, enabling them to deliver robust outcomes across diverse budgets.

## 4 Experiments

### 4.1 Setup

**Backbones** To evaluate the performance of A*-thought for compressing long CoT sequences for modern LRMs, we apply several compression methods to representative reasoning models, including QwQ-32B[3], DeepSeek-R1-Distill-Qwen-32B[4], and s1.1-32B[5].

**Training Data and Verification Model** We utilize the long CoT data released by Muennighoff et al. (2025)[6] as the original CoT data and employ the corresponding distilled model, s1.1-32B, as the verification model, following the approach detailed in Section 3.2.

**Benchmarks** We employ the following mathematical reasoning tasks in our experiments, all of which demand complex reasoning capabilities from LRMs: MATH500 (Lightman et al., 2023), AMC23 (AMC, 2025), OlympiadBench (He et al., 2024), and GSM8K (Cobbe et al., 2021). Model performance is evaluated using the following metrics:

- *Accuracy*: The proportion of model outputs that match the ground-truth answers, measuring the model's correctness.
- *Length*: The average length (i.e., number of tokens) of the model's response; longer responses typically incur higher inference costs.
- *Accuracy per Computation Unit* (ACU) (Ma et al., 2025b): A metric assessing the trade-off between performance and efficiency, calculated as $\mathrm{ACU} = \mathrm{Accuracy}/\mathrm{Length}$.

**Baselines** We compare our method against the following baselines:

- *Chain-of-Draft* (CoD) (Xu et al., 2025): A prompt-based method designed to guide LRMs in generating compact reasoning steps, each typically comprising fewer than five words.
- *Break-the-Chain* (BtC) (Ding et al., 2024): A prompt-based method employing specialized prompting strategies to encourage LRMs to utilize shortcuts, thereby enabling them to rapidly explore reasoning clues while bypassing detailed intermediate steps.
- *TokenSkip* (Xia et al., 2025): A training-based method that first employs prompt compression (Jiang et al., 2024) to shorten long CoT data, and then uses this compressed data to train an efficient reasoning model.

For comparison, we also report the performance of the QwQ-32B model directly fine-tuned on the `s1K-1.1` dataset.

---

[3] `https://huggingface.co/Qwen/QwQ-32B`
[4] `https://huggingface.co/deepseek-ai/DeepSeek-R1-Distill-Qwen-32B`
[5] `https://huggingface.co/simplescaling/s1.1-32B`
[6] `https://huggingface.co/datasets/simplescaling/s1K-1.1`

**Training Details**  We trained all models, including training-based baselines and our proposed method, for 3 epochs with a peak learning rate of $1 \times 10^{-5}$ and a warm-up ratio of $0.1$. Training was conducted on 8 NVIDIA A100 80G GPUs, using a per-GPU batch size of 1 and 8 gradient accumulation steps. For our proposed method, the default hyperparameters were set as $\alpha = 0.5$ (Eq. 4) and $\beta = 0.1$ (Eq. 8). The lower bound for the verification depth, $k_{\min}$, is set to 5, while the upper bound for the search tree depth, $k_{\max}$, is set to 20. The exploration size $W$ was set to 2.

Table 1: Experimental results of different long-to-short methods across several benchmarks. The best results are shown in **bold**, and the second-best results are underlined.

| Methods | MATH500 | | AMC23 | | OlympiadBench | | GSM8K | | Average | | ACU |
|---|---|---|---|---|---|---|---|---|---|---|---|
| | Acc.$^{(\uparrow)}$ | Len.$^{(\downarrow)}$ | Acc.$^{(\uparrow)}$ | Len.$^{(\downarrow)}$ | Acc.$^{(\uparrow)}$ | Len.$^{(\downarrow)}$ | Acc.$^{(\uparrow)}$ | Len.$^{(\downarrow)}$ | Acc.$^{(\uparrow)}$ | Len.$^{(\downarrow)}$ | |
| **Budget: 512 Tokens** | | | | | | | | | | | |
| QwQ-32B | 10.8 | 512.00 | 2.5 | 512.00 | 3.3 | 512.00 | 27.6 | 511.97 | 11.1 | 511.99 | 2.16 |
| QwQ-32B w/ s1K-1.1 | 9.6 | 512.00 | 7.5 | 512.00 | 3.4 | 512.00 | 28.8 | 512.00 | 12.3 | 512.00 | 2.41 |
| + CoD | 10.6 | 512.00 | 5.0 | 512.00 | 4.2 | 512.00 | 29.0 | 511.96 | 12.2 | 511.99 | 2.38 |
| + BtC Effective Shortcut | 10.2 | 512.00 | 12.5 | 512.00 | 4.2 | 512.00 | 26.7 | 511.95 | 13.4 | 511.99 | 2.62 |
| + BtC Skip Steps | 9.6 | 512.00 | 5.0 | 512.00 | 5.6 | 512.00 | 28.9 | 511.95 | 12.3 | 511.99 | 2.40 |
| + TokenSkip | 10.8 | 511.05 | 2.5 | 512.00 | 3.9 | 512.00 | 26.4 | 508.11 | 10.9 | 510.79 | 2.13 |
| + A*-Thought | 33.2 | 491.92 | 15.0 | 508.60 | 12.0 | 509.74 | 57.4 | 451.76 | 29.4 | 490.51 | 5.99 |
| **Budget: 1024 Tokens** | | | | | | | | | | | |
| QwQ-32B | 16.6 | 1016.85 | 15.0 | 1024.00 | 6.4 | 1023.93 | 49.1 | 951.96 | 21.8 | 1004.19 | 2.17 |
| QwQ-32B w/ s1K-1.1 | 24.8 | 1023.52 | 17.5 | 1024.00 | 8.9 | 1023.94 | 60.1 | 999.80 | 27.8 | 1017.82 | 2.73 |
| + CoD | 24.8 | 1023.37 | 5.0 | 1024.00 | 7.3 | 1023.64 | 60.1 | 996.84 | 24.3 | 1016.96 | 2.39 |
| + BtC Effective Shortcut | 23.4 | 1022.88 | 7.5 | 1024.00 | 7.7 | 1023.92 | 61.3 | 1000.44 | 25.0 | 1017.81 | 2.45 |
| + BtC Skip Steps | 23.4 | 1023.25 | 5.0 | 1024.00 | 7.6 | 1024.00 | 59.9 | 1000.93 | 24.0 | 1018.05 | 2.36 |
| + TokenSkip | 22.4 | 995.96 | 12.5 | 1024.00 | 6.4 | 1019.61 | 49.7 | 934.74 | 22.8 | 993.58 | 2.29 |
| + A*-Thought | 50.8 | 858.28 | 37.5 | 928.25 | 22.3 | 954.74 | 81.9 | 688.69 | 48.1 | 857.49 | 5.61 |
| **Budget: 2048 Tokens** | | | | | | | | | | | |
| QwQ-32B | 51.2 | 1844.96 | 25.0 | 1978.60 | 18.4 | 2021.95 | 80.4 | 1245.68 | 43.8 | 1772.80 | 2.47 |
| QwQ-32B w/ s1K-1.1 | 60.0 | 1887.15 | 35.0 | 2000.95 | 23.3 | 2012.14 | 88.7 | 1474.00 | 51.8 | 1843.56 | 2.81 |
| + CoD | 60.2 | 1894.54 | 30.0 | 2022.35 | 25.5 | 2018.02 | 89.5 | 1490.23 | 51.3 | 1856.29 | 2.76 |
| + BtC Effective Shortcut | 60.8 | 1884.67 | 35.0 | 2004.65 | 23.7 | 2012.43 | 89.8 | 1473.25 | 52.3 | 1843.75 | 2.84 |
| + BtC Skip Steps | 58.8 | 1884.96 | 35.0 | 2005.67 | 23.2 | 2013.05 | 89.2 | 1490.39 | 51.6 | 1848.52 | 2.79 |
| + TokenSkip | 53.6 | 1685.34 | 35.0 | 1923.25 | 19.7 | 1943.68 | 86.7 | 1272.03 | 48.8 | 1706.08 | 2.86 |
| + A*-Thought | 69.2 | 1271.76 | 45.0 | 1540.30 | 30.3 | 1625.89 | 91.2 | 843.69 | 58.9 | 1320.41 | 4.46 |
| **Budget: 4096 Tokens** | | | | | | | | | | | |
| QwQ-32B | 75.4 | 2798.67 | 55.0 | 3456.05 | 36.5 | 3645.22 | 85.8 | 1348.24 | 63.2 | 2812.05 | 2.25 |
| QwQ-32B w/ s1K-1.1 | 79.6 | 2693.27 | 65.0 | 3485.95 | 42.4 | 3500.66 | 95.2 | 1624.11 | 70.6 | 2826.00 | 2.50 |
| + CoD | 80.2 | 2719.00 | 60.0 | 3354.28 | 42.0 | 3488.67 | 95.0 | 1655.80 | 69.3 | 2804.44 | 2.47 |
| + BtC Effective Shortcut | 79.6 | 2696.72 | 57.5 | 3355.43 | 42.4 | 3493.28 | 94.8 | 1636.45 | 68.6 | 2795.47 | 2.45 |
| + BtC Skip Steps | 80.2 | 2710.83 | 57.5 | 3399.93 | 41.8 | 3494.41 | 94.9 | 1651.37 | 68.6 | 2814.14 | 2.44 |
| + TokenSkip | 74.4 | 2336.29 | 52.5 | 3156.68 | 37.8 | 3289.44 | 94.8 | 1412.87 | 64.9 | 2548.82 | 2.55 |
| + A*-Thought | 78.8 | 1699.78 | 65.0 | 2385.85 | 40.1 | 2546.45 | 93.1 | 874.54 | 69.3 | 1876.66 | 3.69 |

## 4.2 Main Results

The detailed experimental results, presented in Table 1, yield the following key insights:

**Up to 2.39× accuracy and 2.49× ACU improvements in low-budget scenarios.**  Specifically, across all examined benchmarks, A*-Thought improve the average accuracy of QwQ-32B from 12.3 to 29.4 when the inference budget is constrained to 512 tokens. Concurrently, the ACU score improves from 2.41 to 5.99. Furthermore, in experiments with inference budgets of 1024 and 2048 tokens, A*-Thought consistently attained superior accuracy and the shortest response lengths.

**Up to 33.59% length reduction without substantial accuracy drop in the 4096-token setting.**  For instance, for the QwQ-32B model, A*-Thought decreased the average response length from 2826.00 to 1876.66 tokens. This significant length reduction resulted in only a slight decrease in average accuracy (from 70.6% to 69.3%). Importantly, A*-Thought also attained the highest ACU score in this setting, outperforming both the prompt-based and the training based baselines.

**Compatible with several models, A*-Thought demonstrates generalizability.**  Figure 4 and Figure 5 display the ACU and performance curves on three distinct backbone models: QwQ-32B, R1-Distill-32B, and s1.1-32B.[7] The results demonstrate A*-Thought's effectiveness across these LRMs, where it achieves the highest efficiency and accuracy under various budget conditions.

---

[7]Detailed results are provided in Appendix D.

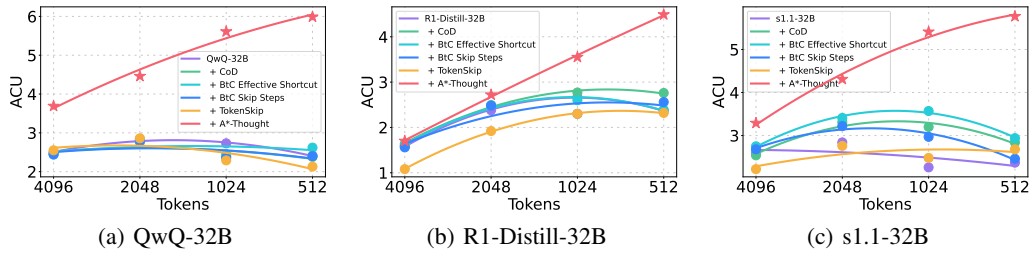

|            |                 |                   |                |
| :--------: | :-------------: | :---------------: | :------------: |
| (a) QwQ-32B | (b) R1-Distill-32B | (c) s1.1-32B |

Figure 4: ACU on different methods, which reflects performance-to-efficiency ratio of LRMs.

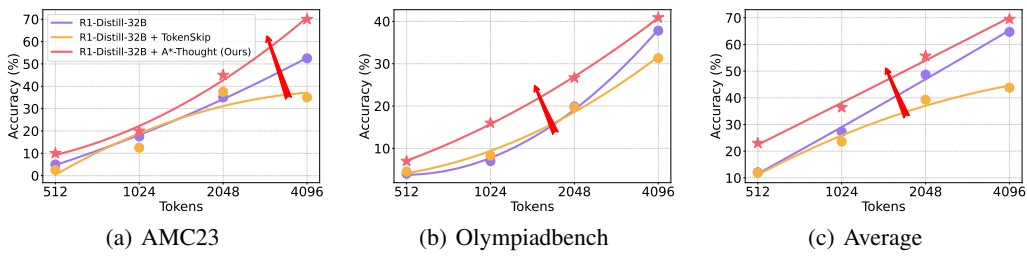

|            |                 |              |
| :--------: | :-------------: | :----------: |
| (a) AMC23 | (b) Olympiadbench | (c) Average |

Figure 5: Performance of R1-Distill-32B augmented using TokenSkip and A*-Thought. "Average" denotes the average accuracy of the model in MATH500, AMC23, OlympiadBench, and GSM8K.

# 5 Analysis

## 5.1 Performance on More Benchmarks

To verify the generalization capabilities of A*-Thought, we conducted supplementary evaluations on the out-of-domain benchmarks LiveCodeBench and MMLU, as shown in Table 2. Although our model was trained exclusively on mathematical tasks, the results demonstrate that it effectively learned the A*-Thought reasoning pattern. This led to improved performance and higher inference efficiency, even on these non-mathematical tasks.

Table 2: Results of A*-Thought on out-of-domain benchmarks.

| Methods | LiveCodeBench | | MMLU | | Average | | ACU |
| :--- | :---: | :---: | :---: | :---: | :---: | :---: | :---: |
| | Acc.$^{(\uparrow)}$ | Len.$^{(\downarrow)}$ | Acc.$^{(\uparrow)}$ | Len.$^{(\downarrow)}$ | Acc.$^{(\uparrow)}$ | Len.$^{(\downarrow)}$ | |
| **Budget: 512 Tokens** | | | | | | | |
| QwQ-32B | 0.0 | 512.00 | 37.6 | 511.94 | 18.8 | 511.97 | 3.67 |
| + A*-Thought | **4.5** | **509.53** | **56.7** | **398.57** | **30.6** | **454.05** | **6.74** |
| **Budget: 1024 Tokens** | | | | | | | |
| QwQ-32B | 0.0 | 1021.94 | 57.4 | 956.90 | 28.7 | 989.42 | 2.90 |
| + A*-Thought | **11.8** | **986.02** | **71.9** | **573.30** | **41.9** | **779.66** | **5.37** |
| **Budget: 2048 Tokens** | | | | | | | |
| QwQ-32B | 3.5 | 1977.58 | 75.2 | 1323.26 | 39.4 | 1650.42 | 2.38 |
| + A*-Thought | **24.5** | **1734.28** | **79.0** | **671.41** | **51.8** | **1202.85** | **4.30** |
| **Budget: 4096 Tokens** | | | | | | | |
| QwQ-32B | 12.0 | 3586.93 | 79.5 | 1584.05 | 45.8 | 2585.49 | 1.77 |
| + A*-Thought | **39.0** | **3044.51** | **80.3** | **733.90** | **59.7** | **1889.21** | **3.16** |

## 5.2 Analysis on the Effect of A* Search

Table 3 presents an ablation study on the value of $k_{\min}$ in the A*-search process. The results show that with $k_{\min} = 15$ and a 4K-token budget, the A*-Thought model consistently outperformed the strongest baseline across all benchmarks. Importantly, this performance was also achieved with higher token efficiency during inference.

Table 3: Effect of the exploration step limit $k_{\min}$ on model performance.

| Methods | MATH500 | | AMC23 | | OlympiadBench | | GSM8K | | Average | | ACU |
|---|---|---|---|---|---|---|---|---|---|---|---|
| | Acc.$^{(\uparrow)}$ | Len.$^{(\downarrow)}$ | Acc.$^{(\uparrow)}$ | Len.$^{(\downarrow)}$ | Acc.$^{(\uparrow)}$ | Len.$^{(\downarrow)}$ | Acc.$^{(\uparrow)}$ | Len.$^{(\downarrow)}$ | Acc.$^{(\uparrow)}$ | Len.$^{(\downarrow)}$ | |
| **Budget: 4096 Tokens** | | | | | | | | | | | |
| QwQ-32B | 75.4 | 2798.67 | 55.0 | 3456.05 | 36.5 | 3645.22 | 85.8 | 1348.24 | 63.2 | 2812.05 | 2.25 |
| QwQ-32B w/ s1K-1.1 | 79.6 | 2693.27 | 65.0 | 3485.95 | 42.4 | 3500.66 | 95.2 | 1624.11 | 70.6 | 2826.00 | 2.50 |
| + A*-Thought $k_{\min} = 5$ | 78.8 | **1699.78** | 65.0 | **2385.85** | 40.1 | **2546.45** | 93.1 | **874.54** | 69.3 | **1876.66** | **3.69** |
| + A*-Thought $k_{\min} = 15$ | **80.8** | 2184.34 | **67.5** | 2893.68 | **44.2** | 3063.96 | **95.5** | 1229.40 | **72.0** | 2342.85 | 3.07 |

## 5.3 Training Loss and Training Time

Figure 6 illustrates the training loss for various training-based methods, demonstrating that A*-Thought achieves the lowest loss. This potential advantage may be attributed to A*-Thought's utilization of the span around individual thoughts, thereby reducing the negative impact of interrupting the complete thinking process. Furthermore, by considering the quality of the intermediate path during the A* search process, we ensure the learnability of the compressed data.

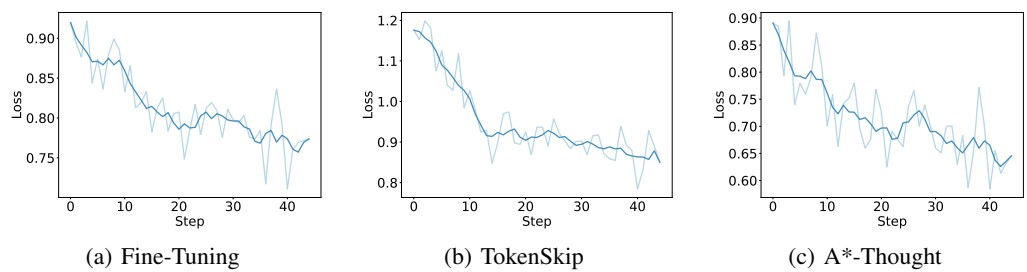

(a) Fine-Tuning      (b) TokenSkip      (c) A*-Thought

Figure 6: Training loss of the training-based methods discussed, using the QwQ-32B backbone.

Table 4 presents a comparison of compressed data sizes and their corresponding training times. Notably, A*-Thought exhibits a significantly higher compression ratio than TokenSkip, while achieving a lower training loss. These results further corroborate the efficacy of the proposed A*-Thought method. A more detailed ablation study on the components of A*-Thought, along with an analysis of the effect of hyperparameters, is presented in the Appendix.

Table 4: Amount of the training data and the corresponding time. $\rho$ denotes the compression rate.

| Training Set | $\rho$ | Training Time (s) | | |
|---|---|---|---|---|
| | | QwQ-32B | R1-Distill-32B | s1.1-32B |
| s1K-1.1 | 100.00% | 13819.60 | 13784.32 | 13638.07 |
| + TokenSkip | 85.41% ($\downarrow$14.59%) | 12846.14 ($\downarrow$7.04%) | 12700.96 ($\downarrow$7.86%) | 12679.52 ($\downarrow$7.03%) |
| + A*-Thought | 31.31% ($\downarrow$68.69%) | 10468.20 ($\downarrow$24.25%) | 10809.50 ($\downarrow$21.58%) | 10550.03 ($\downarrow$22.64%) |

## 5.4 Case Study

Figure 7 showcases a representative example of outputs from models trained with and without A*-Thought, given an identical question. While both models arrive at the correct solution, A*-Thought fosters a more streamlined and focused thinking path. Appendix G provides further comprehensive examples. These examples demonstrate that A*-Thought can train models to adopt a more concise thinking process. We believe this approach, with additional effort and integration with advanced training techniques, holds the potential to break through the current bottlenecks in LRMs.

**Figure 7:** Responses generated by QwQ-32B models trained with and without A*-Thought. Red box represents the question, purple box represents the thinking process, blue box represents the solution.

# 6 Related Works

Recent LRMs achieve complex reasoning capabilities, often described as "deep thinking", through significant computational effort during inference, a process sometimes referred to as test-time scaling. However, studies indicate that models utilizing long CoT processes can be susceptible to excessive computation, leading to inefficiencies often termed "overthinking". Consequently, considerable research has aimed to improve reasoning efficiency. Primary strategies include the development of refined prompting techniques (Renze & Guven, 2024; Ding et al., 2024; Han et al., 2025; Lee et al., 2025a; Aytes et al., 2025; Xu et al., 2025; Ma et al., 2025a) and methods for CoT compression (Hao et al., 2024; Cheng & Durme, 2024; Shen et al., 2025; Zhang et al., 2025; Su et al., 2025).

Specific approaches have targeted token-level or path-level optimizations. For example, Token-Skip (Xia et al., 2025) attempts to focus computation by selectively processing tokens deemed most relevant to the input query while omitting less pertinent ones. Another method, Retro-Search (Lu et al., 2025), inspired by Monte Carlo Tree Search, explores numerous potential reasoning paths within long CoT structures. Its goal is to mitigate redundancy across multiple paths (overthinking) while ensuring adequate exploration within individual paths (avoiding underthinking), issues potentially exacerbated by frequent shifts in reasoning strategy.

In contrast to methods focusing on specific causes of redundancy within long CoT, this paper proposes a heuristic approach. We employ a bidirectional importance score to guide the A* search algorithm. The objective is to identify a computationally efficient yet effective reasoning pathway, thereby contributing to the development of more resource-efficient LRMs.

# 7 Conclusion

We introduce A*-Thought, an innovative CoT compression algorithm designed to enhance the reasoning efficiency of contemporary LRMs. Our approach meticulously crafts concise reasoning pathways. This is achieved by first evaluating the necessity of individual reasoning steps using a novel bidirectional importance score. Subsequently, the thinking steps are assembled into a compact and coherent reasoning chain by employing a path-level A* search algorithm. A key aspect of our A* search process is the design of specialized cost functions. These functions assess not only the quality of the current partial reasoning path but also estimate the potential cost to complete the thought process, thereby providing helpful guidance for selecting the most promising reasoning trajectories. Experimental results demonstrate that A*-Thought can outperform several representative baselines.

## Acknowledgments and Disclosure of Funding

This study is sponsored by CAAI-Lenovo Blue Sky Research Fund. This work is also supported by the AI9Stars community.

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

# A  A*-Thought Algorithm Detail

Algorithm 1 shows the details of the A* search algorithm to compress lengthy CoTs.

---

**Algorithm 1** A*-Thought algorithm for compressing lengthy CoTs

---

1: **Input:**
2:    $\mathbf{q}$: question; $\mathbf{t}$: original CoT; $\mathbf{t}_{\text{sort}}$: thought list sorted by BIS; $\mathbf{s}$: solution; $\mathcal{V}$: verification model; $k_{\min}$: min verification depth; $k_{\max}$: max search depth; $W$: number of observable nodes
3: **Output:**
4:    $\mathbf{t}' \subseteq \mathbf{t}$: compressed thought path
5: **procedure** A* SEARCH
                                           ▷ (1) Initialization
6:        $\mathcal{Q} \leftarrow \mathbf{t}_{\text{sort}}$
7:        $\mathbf{t}'_k \leftarrow \mathcal{Q}.pop()$
8:        $k = 1$
9:        **while** $\mathcal{Q}$ not empty **do**
10:            **if** $k \geq k_{\max}$ **then**
11:                **return** $\mathbf{t}' = \mathbf{t}$
12:            **end if**
                                           ▷ (2) Verification
13:            **if** $k \geq k_{\min}$ **and** $\mathcal{V}(\mathbf{q} + \mathbf{t}'_k) == \mathbf{s}$ **then**
14:                **return** $\mathbf{t}' = \mathbf{t}'_k$
15:            **end if**
                                           ▷ (3) Exploration
16:            $\{\mathbf{r}_1, \ldots, \mathbf{r}_W\} \leftarrow$ first $W$ elements in $\mathcal{Q}$
17:            **for** $\mathbf{r}_w$ in $\{\mathbf{r}_1, \ldots, \mathbf{r}_W\}$ **do**
18:                $f(\mathbf{t}'_k + \mathbf{r}_w) = g(\mathbf{t}'_k + \mathbf{r}_w) + h(\mathbf{t}'_k + \mathbf{r}_w)$
19:            **end for**
20:            $\hat{\mathbf{r}}_w = \operatorname{argmin}_{w \in \{1, \cdots, W\}} f(\mathbf{t}'_k + \mathbf{r}_w)$
21:            $\mathbf{t}'_{k+1} = \langle \mathbf{t}'_k, \hat{\mathbf{r}}_w \rangle$
22:            $\mathcal{Q}.pop(\hat{\mathbf{r}}_w)$
23:            $k = k + 1$
24:        **end while**
25:        **return** $\mathbf{t}'_k$
26: **end procedure**

---

# B    Test-Time-Scaling Strategies

We ran the A*-Thought evaluation on MATH500 and LiveCodeBench, this time using a best-of-N strategy (with N=8) to calculate pass@k. The findings indicate that A*-Thought and test-time-scaling strategies can be used in combination, the experiment results are shown in Table 5.

Table 5: Experimental results on A*-Thought with test-time-scaling strategies.

| Methods | MATH500 | | | | LiveCodeBench | | | |
|---|---|---|---|---|---|---|---|---|
| | pass@1 | pass@2 | pass@4 | pass@8 | pass@1 | pass@2 | pass@4 | pass@8 |
| **Budget: 512 Tokens** | | | | | | | | |
| QwQ-32B | 10.8 | 17.2 | 25.9 | 34.4 | 0.0 | 0.0 | 0.0 | 0.0 |
| + A*-Thought | **33.2** | **47.4** | **60.7** | **70.2** | **4.5** | **7.0** | **12.5** | **20.8** |
| **Budget: 1024 Tokens** | | | | | | | | |
| QwQ-32B | 16.6 | 29.6 | 42.1 | 53.2 | 0.0 | 0.1 | 0.1 | 0.3 |
| + A*-Thought | **50.8** | **65.4** | **75.8** | **81.8** | **11.8** | **20.4** | **31.9** | **44.8** |
| **Budget: 2048 Tokens** | | | | | | | | |
| QwQ-32B | 51.2 | 59.8 | 67.8 | 74.8 | 2.1 | 3.4 | 5.1 | 7.0 |
| + A*-Thought | **69.2** | **79.2** | **85.4** | **89.0** | **24.5** | **34.0** | **45.2** | **56.5** |
| **Budget: 4096 Tokens** | | | | | | | | |
| QwQ-32B | 75.4 | 80.2 | 84.0 | 86.8 | 12.4 | 17.7 | 23.0 | 28.0 |
| + A*-Thought | **78.8** | **86.8** | **91.1** | **93.2** | **39.0** | **50.9** | **60.9** | **68.8** |

# C    Performance of Models of Varying Sizes

This section reports experiments on Qwen2.5-7B-Instruct and Qwen2.5-14B-Instruct in Table 6.

Table 6: Experimental results on models of varying sizes.

| Methods | MATH500 | | AMC23 | | OlympiadBench | | GSM8K | | Average | | ACU |
|---|---|---|---|---|---|---|---|---|---|---|---|
| | Acc.$^{(\uparrow)}$ | Len.$^{(\downarrow)}$ | Acc.$^{(\uparrow)}$ | Len.$^{(\downarrow)}$ | Acc.$^{(\uparrow)}$ | Len.$^{(\downarrow)}$ | Acc.$^{(\uparrow)}$ | Len.$^{(\downarrow)}$ | Acc.$^{(\uparrow)}$ | Len.$^{(\downarrow)}$ | |
| **Budget: 512 Tokens** | | | | | | | | | | | |
| TokenSkip-7B | 9.4 | 511.41 | 2.5 | 512.00 | 4.0 | 512.00 | 25.5 | 508.40 | 10.4 | 511.0 | 2.03 |
| A*-Thought-7B | **30.8** | **499.12** | **5.0** | **508.70** | **6.7** | **509.36** | **59.2** | **447.22** | **25.4** | **491.10** | **5.18** |
| TokenSkip-14B | 9.2 | 510.89 | 7.5 | 512.00 | 4.8 | 512.00 | 29.0 | 511.17 | 12.6 | 511.52 | 2.47 |
| A*-Thought-14B | **30.4** | **497.54** | **10.0** | **512.00** | **10.4** | **508.77** | **50.9** | **475.30** | **25.4** | **498.40** | **5.10** |
| **Budget: 1024 Tokens** | | | | | | | | | | | |
| TokenSkip-7B | 24.8 | 986.62 | 10.0 | 1021.33 | 7.9 | 1019.11 | 51.0 | 907.40 | 23.4 | 983.62 | 2.38 |
| A*-Thought-7B | **48.2** | **866.93** | **25.0** | **991.42** | **19.3** | **961.42** | **74.3** | **661.99** | **41.7** | **870.44** | **4.79** |
| TokenSkip-14B | 23.6 | 991.65 | 10.0 | 1024.00 | 7.4 | 1020.22 | 47.8 | 960.13 | 22.2 | 999.00 | 2.22 |
| A*-Thought-14B | **49.2** | **873.53** | **22.5** | **997.33** | **18.8** | **966.55** | **78.3** | **738.61** | **42.2** | **894.01** | **4.72** |
| **Budget: 2048 Tokens** | | | | | | | | | | | |
| TokenSkip-7B | 35.4 | 1776.50 | 20.0 | 1968.47 | 14.7 | 1959.13 | 55.6 | 1534.37 | 31.4 | 1809.62 | 1.74 |
| A*-Thought-7B | **59.2** | **1352.54** | **30.0** | **1715.22** | **25.4** | **1677.38** | **77.6** | **874.11** | **48.1** | **1404.81** | **3.42** |
| TokenSkip-14B | 55.8 | 1678.03 | 20.0 | 1934.05 | 20.8 | 1940.58 | 73.8 | 1453.71 | 42.6 | 1751.59 | 2.43 |
| A*-Thought-14B | **64.2** | **1308.18** | **40.0** | **1650.40** | **28.3** | **1610.04** | **86.1** | **930.89** | **54.7** | **1374.88** | **3.97** |
| **Budget: 4096 Tokens** | | | | | | | | | | | |
| TokenSkip-7B | 39.8 | 3201.90 | 27.5 | 3593.93 | 20.9 | 3566.00 | 63.4 | 2651.40 | 37.9 | 3253.31 | 1.16 |
| A*-Thought-7B | **61.8** | **1999.10** | **30.0** | **2840.70** | **29.6** | **3048.73** | **78.0** | **1271.18** | **49.9** | **2289.93** | **2.18** |
| TokenSkip-14B | 68.4 | 2406.16 | 40.0 | 3338.80 | 32.5 | 3334.73 | 81.2 | 2054.78 | 55.5 | 2783.62 | 1.99 |
| A*-Thought-14B | **70.8** | **1677.00** | **42.5** | **2350.32** | **34.6** | **2494.24** | **88.8** | **1100.52** | **59.2** | **1905.52** | **3.11** |

# D  Detailed Results of R1-Distill-32B and s1.1-32B

This section reports experiments on R1-Distill-32B and s1.1-32B in Table 7 and Table 8, respectively.

Table 7: Experimental results on DeepSeek-R1-Distill-Qwen-32B.

| Methods | MATH500 | | AMC23 | | OlympiadBench | | GSM8K | | Average | | ACU |
|---|---|---|---|---|---|---|---|---|---|---|---|
| | Acc.$^{(\uparrow)}$ | Len.$^{(\downarrow)}$ | Acc.$^{(\uparrow)}$ | Len.$^{(\downarrow)}$ | Acc.$^{(\uparrow)}$ | Len.$^{(\downarrow)}$ | Acc.$^{(\uparrow)}$ | Len.$^{(\downarrow)}$ | Acc.$^{(\uparrow)}$ | Len.$^{(\downarrow)}$ | |
| **Budget: 512 Tokens** | | | | | | | | | | | |
| R1-Distill-32B w/ s1K-1.1 | 11.2 | 512.00 | 5.0 | 512.00 | 4.0 | 512.00 | 28.1 | 512.00 | 12.1 | 512.00 | 2.36 |
| + CoD | 10.0 | 512.00 | 7.5 | 512.00 | 4.0 | 512.00 | 35.0 | 512.00 | 14.1 | 512.00 | 2.76 |
| + BtC Effective Shortcut | 9.6 | 512.00 | 5.0 | 512.00 | 4.0 | 512.00 | 30.3 | 512.00 | 12.2 | 512.00 | 2.39 |
| + BtC Skip Steps | 11.4 | 512.00 | 7.5 | 512.00 | 4.2 | 512.00 | 29.4 | 512.00 | 13.1 | 512.00 | 2.56 |
| + TokenSkip | 12.2 | 512.00 | 2.5 | 512.00 | 4.5 | 512.00 | 28.4 | 512.00 | 11.9 | 512.00 | 2.32 |
| + A*-Thought | **24.0** | 512.00 | **10.0** | 512.00 | **7.0** | 512.00 | **51.0** | 512.00 | **23.0** | 512.00 | **4.49** |
| **Budget: 1024 Tokens** | | | | | | | | | | | |
| R1-Distill-32B w/ s1K-1.1 | 24.2 | 1024.00 | 17.5 | 1024.00 | 7.0 | 1024.00 | 60.5 | 1024.00 | 27.3 | 1024.00 | 2.67 |
| + CoD | 25.8 | 1024.00 | 10.0 | 1024.00 | 8.0 | 1024.00 | 69.8 | 1024.00 | 28.4 | 1024.00 | 2.77 |
| + BtC Effective Shortcut | 24.4 | 1024.00 | 10.0 | 1024.00 | 8.8 | 1024.00 | 63.5 | 1024.00 | 26.7 | 1024.00 | 2.60 |
| + BtC Skip Steps | 19.4 | 1024.00 | 7.5 | 1024.00 | 7.3 | 1024.00 | 60.2 | 1024.00 | 23.6 | 1024.00 | 2.30 |
| + TokenSkip | 19.0 | 1024.00 | 12.5 | 1024.00 | 8.3 | 1024.00 | 54.4 | 1023.74 | 23.6 | 1023.94 | 2.30 |
| + A*-Thought | **39.6** | 1024.00 | **20.0** | 1024.00 | **16.0** | 1024.00 | **69.8** | 1024.00 | **36.4** | 1024.00 | **3.55** |
| **Budget: 2048 Tokens** | | | | | | | | | | | |
| R1-Distill-32B w/ s1K-1.1 | 54.2 | 2048.00 | 35.0 | 2048.00 | 19.9 | 2048.00 | 85.8 | 2048.00 | 48.7 | 2048.00 | 2.38 |
| + CoD | 57.8 | 2048.00 | 35.0 | 2048.00 | 22.4 | 2048.00 | 88.4 | 2048.00 | 50.9 | 2048.00 | 2.49 |
| + BtC Effective Shortcut | 55.8 | 2048.00 | 37.5 | 2048.00 | 23.3 | 2048.00 | 87.6 | 2048.00 | 51.1 | 2048.00 | 2.49 |
| + BtC Skip Steps | 56.2 | 2048.00 | 37.5 | 2048.00 | 21.4 | 2048.00 | 87.9 | 2048.00 | 50.8 | 2048.00 | 2.48 |
| + TokenSkip | 40.8 | 2048.00 | 37.5 | 2048.00 | 19.7 | 2048.00 | 59.1 | 2048.00 | 39.3 | 2048.00 | 1.92 |
| + A*-Thought | **61.8** | 2048.00 | **45.0** | 2048.00 | **26.7** | 2048.00 | **89.4** | 2048.00 | **55.7** | 2048.00 | **2.72** |
| **Budget: 4096 Tokens** | | | | | | | | | | | |
| R1-Distill-32B w/ s1K-1.1 | **77.4** | 4096.00 | 52.5 | 4096.00 | 37.8 | 4096.00 | 90.9 | 4096.00 | 64.7 | 4096.00 | 1.58 |
| + CoD | 75.4 | 4096.00 | 60.0 | 4096.00 | 40.8 | 4096.00 | 91.7 | 4096.00 | 67.0 | 4096.00 | 1.64 |
| + BtC Effective Shortcut | 75.8 | 4096.00 | 55.0 | 4096.00 | 39.8 | 4096.00 | 91.7 | 4096.00 | 65.6 | 4096.00 | 1.60 |
| + BtC Skip Steps | 73.2 | 4096.00 | 52.5 | 4096.00 | 38.6 | 4096.00 | 92.1 | 4096.00 | 64.1 | 4096.00 | 1.56 |
| + TokenSkip | 51.0 | **4014.70** | 35.0 | 4096.00 | 31.3 | **4068.53** | 57.8 | 3969.45 | 43.8 | **4037.17** | 1.08 |
| + A*-Thought | 75.4 | 4096.00 | **70.0** | 4096.00 | **40.9** | 4096.00 | 91.7 | 4096.00 | **69.5** | 4096.00 | **1.70** |

Table 8: Experimental results on s1.1-32B.

| Methods | MATH500 | | AMC23 | | OlympiadBench | | GSM8K | | Average | | ACU |
|---|---|---|---|---|---|---|---|---|---|---|---|
| | Acc.$^{(\uparrow)}$ | Len.$^{(\downarrow)}$ | Acc.$^{(\uparrow)}$ | Len.$^{(\downarrow)}$ | Acc.$^{(\uparrow)}$ | Len.$^{(\downarrow)}$ | Acc.$^{(\uparrow)}$ | Len.$^{(\downarrow)}$ | Acc.$^{(\uparrow)}$ | Len.$^{(\downarrow)}$ | |
| **Budget: 512 Tokens** | | | | | | | | | | | |
| s1.1-32B | 9.8 | 512.00 | 5.0 | 512.00 | 5.5 | 512.00 | 28.3 | 511.85 | 12.2 | 511.96 | 2.37 |
| + CoD | 12.6 | 511.86 | 7.5 | 512.00 | 5.3 | 512.00 | 32.8 | 511.21 | 14.6 | 511.77 | 2.84 |
| + BtC Effective Shortcut | 13.4 | 511.76 | 10.0 | 512.00 | 4.5 | 512.00 | 32.3 | 511.69 | 15.1 | 511.86 | 2.94 |
| + BtC Skip Steps | 12.2 | 512.00 | 2.5 | 512.00 | 4.5 | 512.00 | 31.2 | 511.95 | 12.6 | 511.99 | 2.46 |
| + TokenSkip | 14.0 | 511.23 | 7.5 | 512.00 | 3.9 | 512.00 | 29.4 | 509.62 | 13.7 | 511.21 | 2.68 |
| + A*-Thought | **34.0** | 494.15 | 12.5 | 505.98 | 11.9 | 509.73 | 56.1 | 474.03 | 28.6 | 495.97 | 5.77 |
| **Budget: 1024 Tokens** | | | | | | | | | | | |
| s1.1-32B | 25.6 | 1014.86 | 10.0 | 1024.00 | 8.8 | 1023.93 | 47.5 | 997.19 | 23.0 | 1015.00 | 2.26 |
| + CoD | 37.8 | 1006.51 | 20.0 | 1019.02 | 11.6 | 1021.62 | 58.7 | 957.21 | 32.0 | 1001.09 | 3.20 |
| + BtC Effective Shortcut | 42.8 | 1001.81 | 25.0 | 1024.00 | 13.8 | 1020.92 | 61.3 | 950.63 | 35.7 | 999.34 | 3.57 |
| + BtC Skip Steps | 32.8 | 1011.61 | 15.0 | 1022.65 | 13.4 | 1022.07 | 58.5 | 975.83 | 29.9 | 1008.04 | 2.97 |
| + TokenSkip | 28.4 | 983.36 | 15.0 | 1020.08 | 9.8 | 1019.31 | 45.4 | 955.14 | 24.7 | 994.47 | 2.48 |
| + A*-Thought | **55.4** | 862.14 | 32.5 | 969.05 | 22.3 | 960.90 | 80.9 | 738.52 | 47.8 | 882.65 | 5.41 |
| **Budget: 2048 Tokens** | | | | | | | | | | | |
| s1.1-32B | 63.2 | 1765.50 | 35.0 | 1986.60 | 26.6 | 1978.44 | 80.9 | 1518.36 | 51.4 | 1812.23 | 2.84 |
| + CoD | 69.6 | 1661.53 | 40.0 | 1906.22 | 31.9 | 1933.45 | 84.9 | 1339.1 | 56.6 | 1710.08 | 3.31 |
| + BtC Effective Shortcut | 70.2 | 1634.15 | 42.5 | 1926.10 | 34.4 | 1928.26 | 85.0 | 1313.87 | 58.0 | 1700.60 | 3.41 |
| + BtC Skip Steps | 67.4 | 1699.65 | 40.0 | 1929.10 | 32.5 | 1954.23 | 84.6 | 1383.99 | 56.1 | 1741.74 | 3.22 |
| + TokenSkip | 59.2 | 1636.53 | 37.5 | 1903.90 | 27.3 | 1916.75 | 70.1 | 1586.98 | 48.5 | 1761.04 | 2.76 |
| + A*-Thought | **74.4** | 1229.54 | 37.5 | 1676.65 | 34.4 | 1635.22 | 88.6 | 913.33 | 58.7 | 1363.69 | 4.31 |
| **Budget: 4096 Tokens** | | | | | | | | | | | |
| s1.1-32B | 80.0 | 2437.12 | **62.5** | 3261.12 | 45.1 | 3364.11 | **92.3** | 1762.46 | **70.0** | 2706.20 | 2.59 |
| + CoD | 74.4 | 2701.37 | 55.0 | 3041.45 | 47.9 | 3228.24 | 90.8 | 1582.40 | 67.0 | 2638.37 | 2.54 |
| + BtC Effective Shortcut | 80.0 | 2209.99 | 60.0 | 3132.45 | 49.3 | 3161.29 | 88.8 | 1609.22 | 69.5 | 2528.24 | 2.75 |
| + BtC Skip Steps | 81.2 | 2325.75 | 60.0 | 3169.82 | 46.4 | 3222.67 | 90.4 | 1673.10 | 69.5 | 2597.84 | 2.68 |
| + TokenSkip | 69.2 | 2404.56 | 57.5 | 3057.28 | 41.5 | 3230.59 | 77.7 | 2375.19 | 61.5 | 2766.91 | 2.22 |
| + A*-Thought | 77.8 | **1629.87** | 52.5 | **2744.00** | 42.4 | **2611.72** | 91.9 | **1055.83** | 66.2 | **2010.36** | 3.29 |

# E Additional Ablation Analysis

By default, we report the average accuracy using QwQ-32B over all the examined benchmarks.

## E.1 Analysis on BIS

The design of BIS significantly influences the quality of assessing the importance of individual thought steps. ATTN (attention level importance) and NLL (model level importance) represent distinct measures of reasoning step significance. As demonstrated in Figure 8, manipulating their individual and combined effects on BIS reveals that their joint application is superior for improving the model performance across various budgets.

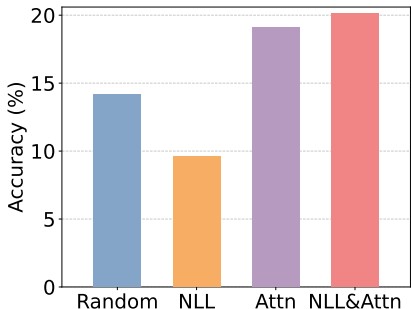

Figure 8: Effect of ATTN and NLL on BIS under the 512-token budget.

The parameter $\alpha$ modulates the balance between question and solution information within BIS. We conducted an ablation analysis, varying $\alpha$ across the discrete set of values [0, 0.25, 0.5, 0.75, 1], to determine its effect on reasoning data quality and, consequently, on model performance. Specifically, with $\alpha = 0$, the score is determined only by the information related to the question. In contrast, setting $\alpha = 1$ results in a score based entirely on information pertaining to the solution. As Figure 9 illustrates, optimal model performance is achieved when the BIS effectively integrates both question and solution perspectives, guided by an appropriate setting of $\alpha$ that ensures a judicious allocation of their respective contributions.

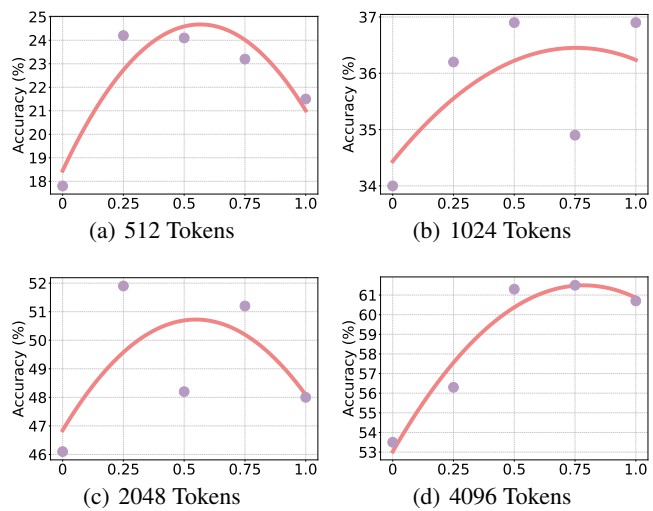

Figure 9: Effect of the hyperparameter $\alpha$ on model performance.

### E.2 Analysis on A* Search

Appropriately setting the maximum exploration steps $k_{\max}$ are keys to optimizing the trade-off between performance and efficiency.

Figure 10 highlights distinct trends: moderate exploration steps are more effective for low-budget scenarios (512-2048 tokens). In contrast, for a 4096-token budget, performance benefits from a greater number of exploration steps. This is likely because more extensive exploration (i.e., deep search) can lead to more concise overall reasoning paths or solutions. Based on these observations, we set $k_{\max} = 20$ in our main experiments by default.

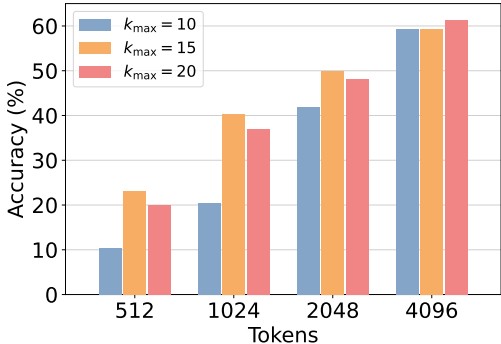

Figure 10: Relationship between the exploration step limit $k_{\max}$ and model performance.

The parameter $\beta$ is used to adjust the weight of the current cost function $g(\cdot)$ in the overall cost function $f(\cdot)$. In the following supplementary experiments, we discussed its discrete values in [0.1, 0.5, 0.9] on the ARC and LiveCodeBench, the experiment results are shown in Table 9.

Table 9: Effect of the hyperparameter $\beta$ on model performance.

| Methods | ARC | | LiveCodeBench | | Average | | ACU |
|---|---|---|---|---|---|---|---|
| | Acc.$^{(\uparrow)}$ | Len.$^{(\downarrow)}$ | Acc.$^{(\uparrow)}$ | Len.$^{(\downarrow)}$ | Acc.$^{(\uparrow)}$ | Len.$^{(\downarrow)}$ | |
| **Budget: 512 Tokens** | | | | | | | |
| $\beta = 0.1$ | **63.5** | **381.02** | 4.5 | 509.53 | **34.00** | **445.28** | **7.64** |
| $\beta = 0.5$ | 52.5 | 438.03 | 4.0 | 510.73 | 28.25 | 474.38 | 5.96 |
| $\beta = 0.9$ | 48.2 | 469.22 | **5.8** | **506.15** | 27.00 | 487.69 | 5.54 |

## F   The Prompt used in this Work

This section details the prompts utilized in this work, including the system prompts presented in Table 10, and the specific CoD (Chain-of-Draft) prompts along with two variants of BtC (Break-the-Chain) baseline prompts, which are shown in Table 11.

Table 10: System prompt

| System Prompt |
| --- |
| Your role as an assistant involves thoroughly exploring questions through a systematic long thinking process before providing the final precise and accurate solutions. This requires engaging in a comprehensive cycle of analysis, summarizing, exploration, reassessment, reflection, backtracing, and iteration to develop well-considered thinking process. 

 Please structure your response into two main sections: Thought and Solution. In the Thought section, detail your reasoning process using the specified format: 

 `<\|begin_of_thought\|>` 
 {thought with steps separated with '\n\n'} 
 `<\|end_of_thought\|>` 

 Each step should include detailed considerations such as analisying questions, summarizing relevant findings, brainstorming new ideas, verifying the accuracy of the current steps, refining any errors, and revisiting previous steps. 

 In the Solution section, based on various attempts, explorations, and reflections from the Thought section, systematically present the final solution that you deem correct. The solution should remain a logical, accurate, concise expression style and detail necessary step needed to reach the conclusion, formatted as follows: 

 `<\|begin_of_solution\|>` 
 {final formatted, precise, and clear solution} 
 `<\|end_of_solution\|>` 

 Now, try to solve the following question through the above guidelines: |

Table 11: Specific prompt for CoD and two variants of BtC

| Methods | Prompt |
| --- | --- |
| CoD | Think step by step, but only keep a minimum draft for each thinking step, with 5 words at most. |
| BtC Effective Shortcut | Rapidly evaluate and use the most effective reasoning shortcut to answer the question. |
| BtC Skip Steps | Let's skip as much as possible. |

## G   Example of the A*-Thought Trajectories

This section provides two comparative examples that illustrate the CoT produced by A*-Thought versus the original CoT of QwQ-32B and s1.1-32B. Figures 11 and 12 demonstrate that A*-Thought successfully reduces redundant thought trajectories while preserving reasoning ability.

Convert the point $(0, 3)$ in rectangular coordinates to polar coordinates. Enter your answer in the form $(r, \theta)$, where $r > 0$ and $0 \leq \theta < 2\pi$.

QwQ-32B

Okay, so I need to convert the rectangular coordinates $(0, 3)$ to polar coordinates. Hmm, let me remember how to do this. Polar coordinates are given as $(r, \theta)$, where $r$ is the distance from the origin to the point, and $\theta$ is the angle made with the positive $x$-axis.

First, to find $r$, the formula is $r = \sqrt{x^2 + y^2}$. In this case, $x$ is 0 and $y$ is 3. Plugging those into the formula, I get $r = \sqrt{0^2 + 3^2} = \sqrt{0 + 9} = \sqrt{9} = 3$. Okay, so $r$ is 3. That seems straightforward.

Now, for the angle $\theta$. The formula for $\theta$ is usually $\theta = \arctan\left(\frac{y}{x}\right)$. But wait, here $x$ is 0. That would make the denominator zero, which would make the tangent undefined. So, $\arctan\left(\frac{y}{x}\right)$ would be $\arctan\left(\frac{3}{0}\right)$, which is undefined. But arctangent of infinity is $\frac{\pi}{2}$, right? Because as $x$ approaches 0 from the positive side, $\frac{y}{x}$ approaches positive infinity, so $\arctan$ approaches $\frac{\pi}{2}$. But since $x$ is exactly 0, we have to consider the direction. If $x$ is 0 and $y$ is positive, the point is on the positive $y$-axis, which is $\theta = \frac{\pi}{2}$. If $x$ is 0 and $y$ is negative, it's $\theta = \frac{3\pi}{2}$. In this case, $y$ is 3, which is positive, so $\theta$ should be $\frac{\pi}{2}$.

**But let me double-check.** Sometimes when converting coordinates, especially when $x$ is 0, the angle can be tricky. Let me visualize the point $(0, 3)$. In rectangular coordinates, that's 3 units up along the $y$-axis. So if I imagine the polar coordinate system, the angle $\theta$ is measured from the positive $x$-axis. If I rotate $\frac{\pi}{2}$ radians (which is 90 degrees) from the positive $x$-axis, I end up pointing straight up along the positive $y$-axis. That's exactly where the point $(0, 3)$ is. So $\theta = \frac{\pi}{2}$ makes sense here.

Therefore, combining $r$ and $\theta$, the polar coordinates should be $\left(3, \frac{\pi}{2}\right)$. But let me make sure there's not another way to represent this. Since polar coordinates can have negative $r$ values and angles adjusted by $\pi$, but the problem specifies that $r > 0$ and $0 \leq \theta < 2\pi$. So we need $r$ positive and $\theta$ within that interval. Since $\theta$ is already $\frac{\pi}{2}$, which is between 0 and $2\pi$, and $r$ is 3, which is positive, this should be the correct answer.

**Wait, but just to confirm, what if I tried to use another angle?** For example, could $\theta$ be $\frac{5\pi}{2}$? But $\frac{5\pi}{2}$ is equivalent to $\frac{\pi}{2}$ when you subtract $2\pi$, so it's the same direction. But since $\theta$ is required to be less than $2\pi$, $\frac{5\pi}{2}$ is out of range. So $\frac{\pi}{2}$ is the correct angle here. Also, if I tried to use a negative $r$, like $r = -3$, then $\theta$ would be $\frac{\pi}{2} + \pi = \frac{3\pi}{2}$. But $r$ must be greater than 0, so that's not allowed. Therefore, $\left(3, \frac{\pi}{2}\right)$ is the only valid polar coordinate representation for the point $(0, 3)$ under the given constraints.

I think that's all. Let me recap: the point is on the positive $y$-axis, so the radius is 3, and the angle is $\frac{\pi}{2}$. Yep, that seems right. I can't think of any mistakes here. The formulas for $r$ and $\theta$ both lead to these values, and considering the position of the point confirms it. So the final answer should be $\left(3, \frac{\pi}{2}\right)$.

Final Answer: The polar coordinates of the point $(0, 3)$ are $\boxed{\left(3, \frac{\pi}{2}\right)}$.

A*-Thought-QwQ-32B

So, in this case, $r$ is 3. Then $\theta$ is the angle made with the positive $x$-axis. The point is $(0, 3)$, which is on the positive $y$-axis. So, the angle $\theta$ here is $\frac{\pi}{2}$ radians, because that's 90 degrees, pointing straight up along the $y$-axis.

**But let me verify.** The formulas for converting from rectangular to polar coordinates are $r = \sqrt{x^2 + y^2}$, which gives $r = \sqrt{0^2 + 3^2} = \sqrt{9} = 3$. Then $\theta$ is $\arctan\left(\frac{y}{x}\right)$. But wait, $x$ is 0 here. So, $\arctan\left(\frac{y}{x}\right)$ would be $\arctan\left(\frac{3}{0}\right)$. However, division by zero is undefined, but in the context of polar coordinates, when $x$ is 0, $\theta$ is either $\frac{\pi}{2}$ or $-\frac{\pi}{2}$, depending on the sign of $y$. Since $y$ is positive (3), $\theta$ is $\frac{\pi}{2}$. So, that checks out. **But let me make sure there isn't another angle that could represent the same point.** For example, sometimes adding $2\pi$ to $\theta$ gives the same direction, but since the problem requires $\theta$ to be between 0 and $2\pi$, $\frac{\pi}{2}$ is already within that range. So, the answer should be $\left(3, \frac{\pi}{2}\right)$.

Final Answer: The polar coordinates are $\boxed{\left(3, \frac{\pi}{2}\right)}$.

To convert the point $(0, 3)$ from rectangular coordinates to polar coordinates, we use the formulas $r = \sqrt{x^2 + y^2}$ and $\theta = \arctan\left(\frac{y}{x}\right)$.

1. Calculate $r$:

$$r = \sqrt{0^2 + 3^2} = \sqrt{9} = 3$$

2. Determine $\theta$:

- Since $x = 0$ and $y = 3$, the point lies on the positive $y$-axis.
- The angle $\theta$ for the positive $y$-axis is $\frac{\pi}{2}$.

Thus, the polar coordinates are $\boxed{\left(3, \frac{\pi}{2}\right)}$.

To convert the rectangular coordinates $(0, 3)$ to polar coordinates, we start by calculating the radius $r$ using the formula $r = \sqrt{x^2 + y^2}$. Here, $x = 0$ and $y = 3$:

$$r = \sqrt{0^2 + 3^2} = \sqrt{9} = 3$$

Next, we determine the angle $\theta$. Since $x = 0$ and $y = 3$ is positive, the point lies on the positive $y$-axis. The angle $\theta$ for a point on the positive $y$-axis is $\frac{\pi}{2}$.

We verify that $\theta = \frac{\pi}{2}$ is within the required interval $0 \leq \theta < 2\pi$ and that $r = 3$ is positive. Thus, the polar coordinates are:

$$\boxed{\left(3, \frac{\pi}{2}\right)}$$

Figure 11: A specific example comparing QwQ-32B and A*-Thought-QwQ-32B (Ours), which red box represents the question, purple box represents the thought path, blue box represents the solution.

What is the value of $x$ such that $\log_2\left(\log_2(2x-2)\right) = 2$ ?

### s1.1-32B

Okay, so I need to find the value of x that satisfies the equation $\log_2(\log_2(2x-2)) = 2$. Hmm, nested logarithms, that could be a bit tricky. Let me take it step by step.

First, let me recall that if I have $\log_a(\log_b(\text{expression})) = c$, then I need to solve the inner logarithm first. But maybe I can work from the outside in. Since the outer logarithm is base 2 and equals 2, that means that the argument of the outer log must be $2^2$. Because $\log_2(y) = 2$ implies $y = 2^2 = 4$. So, the inner logarithm must equal 4. Wait, let me confirm that. If $\log_2(\text{something}) = 2$, then that something must be $2^2$, which is 4. So, the equation simplifies to $\log_2(2x-2) = 4$. Then, solving for the inner logarithm, $\log_2(2x-2) = 4$ implies that $2x-2$ must be $2^4$, which is 16. Therefore, $2x-2 = 16$. Solving for x, add 2 to both sides: $2x = 18$, then divide by 2: $x = 9$. So, $x = 9$.

**But wait, I need to check if this value of x actually satisfies the original equation.** Let's plug $x = 9$ back into the original equation and see.

Original equation: $\log_2(\log_2(2x-2)) = 2$.

First, compute the inner expression: $2x-2 = 2 \times 9 - 2 = 18 - 2 = 16$. Then, $\log_2(16) = 4$, because $2^4 = 16$. Then, $\log_2(4) = 2$, since $2^2 = 4$. Therefore, the original equation becomes $\log_2(4) = 2$, which is correct. So, $x = 9$ works.

**But wait, I should also check if there are any domain restrictions.** The argument of the inner logarithm, $2x-2$, must be positive because you can't take the logarithm of a non-positive number. So, $2x-2 > 0$. Solving for x: $2x > 2$, so $x > 1$. Since $x = 9$ is greater than 1, it's valid. Also, the argument of the outer logarithm is $\log_2(2x-2)$, which must be positive because the logarithm is only defined for positive numbers. Wait, but the equation is $\log_2(\log_2(2x-2)) = 2$. For $\log_2(\log_2(2x-2))$ to be defined, $\log_2(2x-2)$ must be positive. Because the logarithm function is only defined for positive arguments. So, $\log_2(2x-2) > 0$. Since $\log_2(y) > 0$ when $y > 1$, this implies that $2x-2 > 1$. So, $2x > 3$, hence $x > 3/2$. So, the domain is $x > 3/2$. Since $x = 9$ is greater than $3/2$, it's valid.

Therefore, the solution is $x = 9$. But let's verify once again. Let's compute the inner expression: $2x-2 = 2 \times 9 - 2 = 16$. Then, $\log_2(16) = 4$. Then, $\log_2(4) = 2$, which matches the original equation. So, it's correct.

**Wait, just to make sure, let me check if there's any other possible solution.** The equation $\log_2(\log_2(2x-2)) = 2$. Let me think. The function $\log_2$ is a strictly increasing function, so the equation $\log_2(y) = 2$ has only one solution $y = 4$. Then, the equation $\log_2(2x-2) = 4$ has only one solution $2x-2 = 16$, leading to $x = 9$. So, there's only one solution.

**Wait, but let me check if there's any extraneous solutions.** When solving equations with logarithms, sometimes you can get extraneous solutions if you squared both sides or something. But in this case, we just used the definition of logarithms and solved linear equations. So, $x = 9$ is the only solution and it satisfies the domain restrictions. Therefore, the answer is $x = 9$.

**Final Answer** $\boxed{9}$

To solve the equation $\log_2\left(\log_2(2x-2)\right) = 2$, we start by recognizing that the outer logarithm must equal 2, which implies that the argument of the outer logarithm must be $2^2 = 4$. Therefore, we set up the equation: $\log_2(2x-2) = 4$ Next, we solve for the inner logarithm. The equation $\log_2(2x-2) = 4$ implies that $2x-2 = 2^4 = 16$. Solving for $x$: $2x-2 = 16 \implies 2x = 18 \implies x = 9$ We verify the solution by substituting $x = 9$ back into the original equation: Compute the inner expression: $2 \cdot 9 - 2 = 16$, compute the inner logarithm: $\log_2(16) = 4$, compute the outer logarithm: $\log_2(4) = 2$. The original equation is satisfied, confirming that $x = 9$ is indeed the solution. Additionally, we check the domain restrictions to ensure that the arguments of the logarithms are positive. The inner argument $2x-2$ must be positive, which requires $x > 1$, and the argument of the outer logarithm, $\log_2(2x-2)$, must also be positive, requiring $x > \frac{3}{2}$. Since $x = 9$ satisfies both conditions, it is valid. Thus, the value of $x$ is $\boxed{9}$.

### A*-Thought-s1.1-32B

So:
$\log_2(2x-2) = 4$
Therefore:
$2x-2 = 16$
Solving for x:
$2x = 16 + 2 = 18$
$x = 18/2 = 9$
So $x = 9$.

**But let's check if this satisfies the original equation.**
Original equation:
$\log_2(\log_2(2x-2)) = 2$
Plug in $x = 9$:
$\log_2(\log_2(2 \times 9 - 2)) = \log_2(\log_2(18-2)) = \log_2(\log_2(16))$.
$\log_2(16) = 4$, because $2^4 = 16$.
So:
$\log_2(4) = 2$, which is correct.
Therefore, $x = 9$ is the correct solution.
**Final Answer** $\boxed{9}$

To solve the equation $\log_2(\log_2(2x-2)) = 2$, we start by isolating the inner logarithm. We know that if $\log_2(y) = 2$, then $y = 2^2 = 4$. Therefore, we can rewrite the equation as: $\log_2(2x-2) = 4$. Next, we solve for $2x-2$ by converting the logarithmic equation to its exponential form. Since $\log_2(2x-2) = 4$, it follows that: $2x-2 = 2^4 = 16$. Solving for $x$, we add 2 to both sides: $2x = 18$. Then, divide by 2: $x = 9$. To verify, we substitute $x = 9$ back into the original equation:

$$\log_2(\log_2(2 \cdot 9 - 2)) = \log_2(\log_2(16)) = \log_2(4) = 2$$

The solution satisfies the original equation, confirming that the value of $x$ is $\boxed{9}$.

Figure 12: A specific example comparing s1.1-32B and A*-Thought-s1.1-32B (Ours), which red box represents the question, purple box represents the thought path, blue box represents the solution.

## H  Limitation and Future Works

At present, the application of A*-Thought is confined to supervised fine-tuning (SFT). However, in light of recent studies showcasing the efficacy of reinforcement learning (RL) in advancing the reasoning capacities of LRMs, we intend to investigate the extension of our methodology to an RL-based approach.

## I  Impact Statement

The profound thinking abilities of LRMs allow them to master complex scenarios, but at the cost of significant computing resource consumption. Our work focuses on developing more efficient and green AI, which promises to decrease the energy footprint of LRM deployment and, crucially, enable their application in more resource-scarce environments like endpoint devices.

