# OpenReview forum: "A*-Thought: Efficient Reasoning via Bidirectional Compression for Low-Resource Settings"
_NeurIPS.cc/2025/Conference — NeurIPS 2025 poster_

### Official Review · Reviewer_JDQU · 2025-06-23

**Clarity:** 3
**Significance:** 3
**Originality:** 3
**Rating:** 4
**Confidence:** 4

**Summary:**

The paper introduces A*-Thought, a novel framework designed to enhance the reasoning efficiency of Large Reasoning Models (LRMs) by compressing their lengthy reasoning chains. The authors propose a bidirectional importance estimation mechanism to evaluate the significance of each reasoning step based on its relevance to both the question and the solution. This mechanism assigns a bidirectional importance score (BIS) to each step, which is then used in conjunction with the A* search algorithm to identify and isolate the most essential thoughts from the extensive reasoning chains. The A* search algorithm employs cost functions that assess the quality of the current path and estimate the future cost to reach a desirable final solution, thereby efficiently compressing the chain of thought and determining a reasoning path with high information density and low cost. Extensive experiments on various advanced math tasks demonstrate that A*-Thought effectively balances performance and efficiency over a huge search space. For instance, it can improve the performance of QwQ-32B by 2.39× with low-budget and reduce the length of the output token by nearly 50% with high-budget. The proposed method is also compatible with several other LRMs, showcasing its generalization capability.

**Questions:**

### Questions

1. What validation model is used during the A* search process? The paper does not seem to specify this, which raises concerns about reproducibility and implementation clarity.

2. The teaser figure is difficult to interpret. What do the circles of different sizes and depths represent? A legend or further explanation would greatly improve readability.

3. Does using `\n\n` as a delimiter for segmenting thought steps risk omitting essential intermediate reasoning steps, especially in multi-line or complex CoT outputs?

4. Have the authors explored **test-time scaling strategies**, such as self-consistency or diverse sampling? Such evaluations would help better understand the upper-bound performance of the method.

**Ethical Concerns:**

["NO or VERY MINOR ethics concerns only"]

**Final Justification:**

All of my concerns have been addressed. I have updated my score based the rebuttal.

**Limitations:**

No.

**Paper Formatting Concerns:**

No.

**Quality:**

3

**Strengths And Weaknesses:**

### Strengths

1. The paper is novel and presents a well-defined problem formulation. Both the **Step-Level Bidirectional Importance Score** and the **Path-Level A\* Search** are intuitively motivated and thoroughly described.
2. The proposed method achieves strong empirical performance, demonstrating a **2.39× speedup** over the 32B model.
3. The overall structure of the paper is clear and easy to follow.

### Weaknesses

1. It would be helpful to include a more detailed analysis of the **computational cost** incurred by the A* search process when selecting training samples across benchmarks.
2. The paper would benefit from a broader evaluation across **different model scales** to better assess robustness.

---

> ### Author Rebuttal · Authors · 2025-07-30
>
> # For Question 1
>
> The verification model $\mathcal{V}$ is the **`original s1.1-32B model`** (reported in the Training Details paragraph of Section 4.1, line 178).
>
> # For Question 2
>
> We have already explained the meaning of the circles and their color intensity (reported in the description of Figure 1).
>
> To be precise, **`all circles are of a consistent size`**, with each representing a thought step.
>
> **`The color of each circle indicates its importance`**, a value quantified by the Bidirectional Importance Score (BIS) from Section 3.1. Consequently, a darker shade signifies a higher importance score.
>
> # For Question 3
>
> Following prior works such as Retro-Search, ThinkPrune, Length-filtered Vote, and SEAL, we split reasoning steps using "\n\n".
>
> Building on this, we are the first to propose a **`"thinking span" strategy`**. When an individual thought step $\mathbf{t}^{(n)}$ is to be explored, we instead select a span centered around it:$$\mathbf{r}^{(n)}=\langle \mathbf{t}^{(n-1)}, \mathbf{t}^{(n)}, \mathbf{t}^{(n+1)}\rangle,$$where $\mathbf{t}^{(n-1)}$ and $\mathbf{t}^{(n+1)}$ are the immediately preceding and succeeding steps of $\mathbf{t}^{(n)}$. This method allows the A* search process to simultaneously consider both the physical order of the original thought path and the logical, importance-based order of the BIS queue.
>
> # For Question 4
>
> We are grateful to the reviewer for this insightful suggestion. We reran the A*-Thought evaluation on MATH-500 and LiveCodeBench, this time using a **best-of-N strategy** (with N=8) to calculate pass@k. The findings indicate that **`A*-Thought and test-time-scaling strategies can be used in combination`**. These results will be integrated into the final version of the paper.
>
> |       Methods       | MATH-500 |        |        |                 | LiveCodeBench |        |        |        |
> | :-----------------: | -------- | ------ | ------ | --------------- | ------------- | ------ | ------ | ------ |
> |                     | pass@1   | pass@2 | pass@4 | pass@8          | pass@1        | pass@2 | pass@4 | pass@8 |
> |                     |          |        |        | **Budget: 512** |               |        |        |        |
> |       QwQ-32B       | 10.8     | 17.2   | 25.9   | 34.4            | 0.0           | 0.0    | 0.0    | 0.0    |
> | + A*-Thought (Ours) | 33.2     | 47.4   | 60.7   | 70.2            | 4.5           | 7.0    | 12.5   | 20.8   |
> |                     |          |        |        | **Budget: 1K**  |               |        |        |        |
> |       QwQ-32B       | 16.6     | 29.6   | 42.1   | 53.2            | 0.0           | 0.1    | 0.1    | 0.3    |
> | + A*-Thought (Ours) | 50.8     | 65.4   | 75.8   | 81.8            | 11.8          | 20.4   | 31.9   | 44.8   |
> |                     |          |        |        | **Budget: 2K**  |               |        |        |        |
> |       QwQ-32B       | 51.2     | 59.8   | 67.8   | 74.8            | 2.1           | 3.4    | 5.1    | 7.0    |
> | + A*-Thought (Ours) | 69.2     | 79.2   | 85.4   | 89.0            | 24.5          | 34.0   | 45.2   | 56.5   |
> |                     |          |        |        | **Budget: 4K**  |               |        |        |        |
> |       QwQ-32B       | 75.4     | 80.2   | 84.0   | 86.8            | 12.4          | 17.7   | 23.0   | 28.0   |
> | + A*-Thought (Ours) | 78.8     | 86.8   | 91.1   | 93.2            | 39.0          | 50.9   | 60.9   | 68.8   |
>
> # For Weakness 1
>
> Our training data is **`independent of the test sets`** (reported in the Training Data and Verification Model paragraph of Section 4.1, lines 177-178).
>
> A*-Thought was run on the 1K training data, this involved a **`5.2-minute`** BIS scoring stage and a **`7.43-hour`** A* search stage. For different benchmarks, the computational cost of our method is then proportional to the length of the output tokens required at inference time.
>
> Crucially, A*-Thought also demonstrates remarkable generalization. Though trained only on math, our supplementary evaluations on LiveCodeBench and MMLU confirm that it successfully applies the A*-Thought pattern to non-math tasks (e.g., coding, history, computer science, law).
>
> |   Methods    | LiveCodeBench |         |      MMLU       |         | Average |         | ACU  |
> | :----------: | :-----------: | :-----: | :-------------: | :-----: | :-----: | :-----: | :--: |
> |              |     Acc.      |  Len.   |      Acc.       |  Len.   |  Acc.   |  Len.   |      |
> |              |               |         | **Budget: 512** |         |         |         |      |
> |   QwQ-32B    |      0.0      | 512.00  |      37.6       | 511.94  |  18.8   | 511.97  | 3.67 |
> | + A*-Thought |      4.5      | 509.53  |      56.7       | 398.57  |  30.6   | 454.05  | 6.74 |
> |              |               |         | **Budget: 1K**  |         |         |         |      |
> |   QwQ-32B    |      0.0      | 1021.94 |      57.4       | 956.90  |  28.7   | 989.42  | 2.90 |
> | + A*-Thought |     11.8      | 986.02  |      71.9       | 573.30  |  41.9   | 779.66  | 5.37 |
> |              |               |         | **Budget: 2K**  |         |         |         |      |
> |   QwQ-32B    |      3.5      | 1977.58 |      75.2       | 1323.26 |  39.4   | 1650.42 | 2.38 |
> | + A*-Thought |     24.5      | 1734.28 |      79.0       | 671.41  |  51.8   | 1202.85 | 4.30 |
> |              |               |         | **Budget: 4K**  |         |         |         |      |
> |   QwQ-32B    |     12.0      | 3586.93 |      79.5       | 1584.05 |  45.8   | 2585.49 | 1.77 |
> | + A*-Thought |     39.0      | 3044.51 |      80.3       | 733.90  |  59.7   | 1889.21 | 3.16 |
>
> # For Weakness 2
>
> We are grateful for the reviewer's insightful suggestion. We conducted new experiments by training the Qwen2.5-**7B**-Instruct and Qwen2.5-**14B**-Instruct on data compressed by A*-Thought and benchmarking their performance against the TokenSkip baseline. The results affirm that A*-Thought is robust and maintains its effectiveness **`across different model sizes`**. These new results will be incorporated into the final manuscript.
>
> |        Methods        |  MATH500 |             |   AMC23  |             |  OlympiadBench  |             |   GSM8K  |             |  Average |             |    ACU   |
> | :-------------------: | :------: | :---------: | :------: | :---------: | :-------------: | :---------: | :------: | :---------: | :------: | :---------: | :------: |
> |                       |   Acc.   |     Len.    |   Acc.   |     Len.    |      Acc.       |     Len.    |   Acc.   |     Len.    |   Acc.   |     Len.    |          |
> |                       |          |             |          |             | **Budget: 512** |             |          |             |          |             |          |
> |     TokenSkip-7B      |      9.4 |      511.41 |      2.5 |      512.00 |       4.0       |      512.00 |     25.5 |      508.40 |     10.4 |       511.0 |     2.03 |
> | A*-Thought-7B (Ours)  | **30.8** |  **499.12** |  **5.0** |  **508.70** |     **6.7**     |  **509.36** | **59.2** |  **447.22** | **25.4** |  **491.10** | **5.18** |
> |     TokenSkip-14B     |      9.2 |      510.89 |      7.5 |      512.00 |       4.8       |      512.00 |     29.0 |      511.17 |     12.6 |      511.52 |     2.47 |
> | A*-Thought-14B (Ours) | **30.4** |  **497.54** | **10.0** |  **512.00** |    **10.4**     |  **508.77** | **50.9** |  **475.30** | **25.4** |  **498.40** | **5.10** |
> |                       |          |             |          |             | **Budget: 1K**  |             |          |             |          |             |          |
> |     TokenSkip-7B      |     24.8 |      986.62 |     10.0 |     1021.33 |       7.9       |     1019.11 |     51.0 |      907.40 |     23.4 |      983.62 |     2.38 |
> | A*-Thought-7B (Ours)  | **48.2** |  **866.93** | **25.0** |  **991.42** |    **19.3**     |  **961.42** | **74.3** |  **661.99** | **41.7** |  **870.44** | **4.79** |
> |     TokenSkip-14B     |     23.6 |      991.65 |     10.0 |     1024.00 |       7.4       |     1020.22 |     47.8 |      960.13 |     22.2 |      999.00 |     2.22 |
> | A*-Thought-14B (Ours) | **49.2** |  **873.53** | **22.5** |  **997.33** |    **18.8**     |  **966.55** | **78.3** |  **738.61** | **42.2** |  **894.01** | **4.72** |
> |                       |          |             |          |             | **Budget: 2K**  |             |          |             |          |             |          |
> |     TokenSkip-7B      |     35.4 |     1776.50 |     20.0 |     1968.47 |      14.7       |     1959.13 |     55.6 |     1534.37 |     31.4 |     1809.62 |     1.74 |
> | A*-Thought-7B (Ours)  | **59.2** | **1352.54** | **30.0** | **1715.22** |    **25.4**     | **1677.38** | **77.6** |  **874.11** | **48.1** | **1404.81** | **3.42** |
> |     TokenSkip-14B     |     55.8 |     1678.03 |     20.0 |     1934.05 |      20.8       |     1940.58 |     73.8 |     1453.71 |     42.6 |     1751.59 |     2.43 |
> | A*-Thought-14B (Ours) | **64.2** | **1308.18** | **40.0** | **1650.40** |    **28.3**     | **1610.04** | **86.1** |  **930.89** | **54.7** | **1374.88** | **3.97** |
> |                       |          |             |          |             | **Budget: 4K**  |             |          |             |          |             |          |
> |     TokenSkip-7B      |     39.8 |     3201.90 |     27.5 |     3593.93 |      20.9       |     3566.00 |     63.4 |     2651.40 |     37.9 |     3253.31 |     1.16 |
> | A*-Thought-7B (Ours)  | **61.8** | **1999.10** | **30.0** | **2840.70** |    **29.6**     | **3048.73** | **78.0** | **1271.18** | **49.9** | **2289.93** | **2.18** |
> |     TokenSkip-14B     |     68.4 |     2406.16 |     40.0 |     3338.80 |      32.5       |     3334.73 |     81.2 |     2054.78 |     55.5 |     2783.62 |     1.99 |
> | A*-Thought-14B (Ours) | **70.8** | **1677.00** | **42.5** | **2350.32** |    **34.6**     | **2494.24** | **88.8** | **1100.52** | **59.2** | **1905.52** | **3.11** |

---

### Official Review · Reviewer_jq25 · 2025-06-30

**Clarity:** 2
**Significance:** 3
**Originality:** 4
**Rating:** 5
**Confidence:** 3

**Summary:**

This paper presents an approach for decreasing the length of chain-of-thought reasoning traces while maintaining accuracy. To do so, they design an algorithm that searches for subtraces that can still reach the final answer, and then conduct supervised fine-tuning (I think?) on those subtraces. Evaluated on mathematical reasoning benchmarks, the resulting models are much more accurate for a given token budget, and maintain accuracy when answering without a budget too. The algorithm for discovering subtraces is said to be a form of A* search that sorts tokens based on their importance in determining the solution, then chooses which tokens to in an expand-then-greedily-select step until the current subtrace is verified to produce a correct solution.

**Questions:**

Q1. Is a reasoning step $t^(n)$ a single token? Or is it like a sentence? If it's not a token, is the splitting of an answer into thoughts given by your dataset? For the rest of my answer, I'll assume it's a single token..
Q1. For the NLL part of the BIS, it looks like $NLL(s | t^(n))$ is the probability of generating the solution given the single reasoning step? Isn't that going to be very hard for the model? Similarly, it seems odd to calculate the $NLL(t^(n) | q)$. But I guess even though it's weird, there's enough signal there to be useful?
Q3. I don't understand the $A^*$ search algorithm. Is this even A^*? Here is my understanding:
    1. sort tokens by BIS score. start with empty trace T.
    2. take the top W tokens. For each one, score using cost function f. Take the best scoring token and add it to the trace.
    3. check if current trace leads to solution using verifier. If it does, return current trace. Otherwise, repeat step 2.

Questions about this algorithm:
Q3.1: why are there two different scoring functions, BIS and the cost function? It seems like BIS more efficient to evaluate, and f more expensive? Why not just use the same exact function for the two, but use a cheap model for the initial sorting function and then the full model for the cost function?
Q3.2: what is the value of W? is this stated somewhere?
Q3.3: the tokens after being sorted by BIS are out of order, right? when added to the trace, they are put back into their order, right?
Q3.4: after expanding W options, are the ones that aren't finally selected put back into the queue, or discarded?
Q4. how does the verifier work? You just say which model it is, but I don't understand what the model is doing.

**Ethical Concerns:**

["NO or VERY MINOR ethics concerns only"]

**Final Justification:**

The authors did a good job explaining details of the approach that I was confused about. I will update my score to a 5.

**Quality:**

3

**Strengths And Weaknesses:**

## Strengths
The approach works really well compared to baselines.
The paper is well-written. Most of it is easy to understand and clear.
The approach is interesting and novel. At first when I read the paper, I was skeptical that such an approach would lead to useful results, so it's surprising to see that it works.
The interesting part of the approach is the fine-grained importance score as the basis of discovering useful traces. It's based on attention weights and the likelihood of the model producing the solution given the tokens up to that point, but using GPT-2 instead of whatever model is generating the reasoning traces.
I find it quite surprising that GPT-2 could yield useful scores this way, given how weak of a model it is.

## Weaknesses
The A* algorithm and the bidirectional score could be explained more clearly. See questions section.
There are some other confusing parts of the paper, see questions section.
The baselines the approach is being compared to seem quite weak — see appendix E for prompts.

---

> ### Author Rebuttal · Authors · 2025-07-30
>
> # For Q1
>
> Each thought step $\mathbf{t}^{(n)}$ is a **`complete sentence`** (reported in Section 3.1, line 105).
>
> Following prior works such as Retro-Search, ThinkPrune, Length-filtered Vote, and SEAL, we split reasoning steps using `\n\n`. Our process begins by extracting the content between the model-specific identifiers, `<think>` and `</think>`. This content is then segmented by `\n\n` to generate a set of thought-step sentences. Following the subsequent bidirectional importance scoring and A* search phases, we identify a minimal subset of these steps, which forms the compressed reasoning output.
>
> # For Q 2
>
> **$\mathrm{NLL}(\mathbf{y} | \mathbf{x})$ reflects the probability of obtaining $\mathbf{y}$ under the condition of $\mathbf{x}$,**  where $\mathbf{x}$ and $\mathbf{y}$ are distinct sentences.
>
> Given that each thought step $\mathbf{t}^{(n)}$ is a **complete sentence**, we can easily calculate Negative Log-Likelihood (NLL) using language models (e.g., GPT-2):
> $$\mathrm{NLL}(\mathbf{y} | \mathbf{x}) = -\frac{1}{|\mathbf{y}|} \sum_{j=1}^{|\mathbf{y}|} \mathrm{log} P \left (
> y_j | \mathbf{x}, \mathbf{y}_{<j} \right ).$$
> A more informative context $\mathbf{x}$ increases the likelihood of generating the target $\mathbf{y}$. We apply this by treating each thought step $\mathbf{t}^{(n)}$  as the context $\mathbf{x}$ and calculating its relevance to both the question $\mathbf{q}$ and the solution $\mathbf{s}$ (as the target $\mathbf{y}$). This results in two key scores: $\mathrm{NLL}\left( \mathbf{q} | \mathbf{t}^{(n)} \right)$ and $\mathrm{NLL}\left( \mathbf{s} | \mathbf{t}^{(n)} \right)$.
>
> To improve scoring and enhance the distinctiveness of the results, we follow LongLLMLingua by inserting a template phrase to explicitly prompt the causal connection. Specifically, we use:
> - For $\mathrm{NLL}\left( \mathbf{q} | \mathbf{t}^{(n)} \right)$: "We can get the solution to this question in the given thought step."
> - For $\mathrm{NLL}\left( \mathbf{s} | \mathbf{t}^{(n)} \right)$: "We can get this solution in the given thought step."
>
> # For Q 3
>
> A* search is a canonical heuristic algorithm in computer science, which was jointly proposed by Peter Hart, Nils Nilsson and Bertram Raphael in 1968. It is widely applied in the fields of path planning and map navigation in AI systems. It uniquely combines search efficiency with the guarantee of discovering the shortest possible path. The algorithm progresses by calculating a total cost function, $F(n)=G(n)+H(n)$, for every node n it considers. Within this function, $G(n)$ represents the exact cost incurred to travel from the origin to node n, while $H(n)$ represents the future cost—a heuristic estimation of the cost from node n to the final destination. It is by virtue of this heuristic information, $H(n)$, that the A* algorithm can intelligently navigate a vast search space without resorting to an exhaustive exploration, thus ensuring computational efficiency.
>
> Drawing an analogy, we can view the standard Chain-of-Thought (CoT) method, shown in Figure 1(a), as an unguided traversal of all intermediate nodes, which is often a convoluted and verbose approach. Inspired by the principles of A* search, our A*-Thought method (Figure 1(b)) introduces a heuristic to this process. Here, the importance of each node—visualized by its color depth—serves as the heuristic. By integrating this information, A*-Thought efficiently searches for the most concise thought trajectory required to solve a problem, which in turn leads to highly efficient reasoning and inference.
>
> Therefore, the overall pipeline is:
> 1. sort **`sentences`** by BIS score. start with empty trace T.
> 2. take the top W **`sentences`**, and respectively obtain the **`thinking span`** centered on the sentence and carrying the preceding and following sentences. For each one, score using cost function F. Take the best scoring **`thinking span`** and add it to the trace.
> 3. check if current trace leads to solution using verifier. If it does, return current trace. Otherwise, repeat step 2.
>
> # For Q 3.1
>
> Both the Bidirectional Importance Score (BIS) and the Cost Function serve to guide the expansion of the search tree, but at different granularities: BIS determines the **coarse-grained global direction**, while the Cost Function determines the **fine-grained local direction**.
> - **Global Direction (BIS)**: In classic pathfinding, the positions of all nodes between the start and end points are fixed, which allows for a straightforward determination of candidate node priority at each search depth. In contrast, for the many thought steps in a Chain-of-Thought, we must first assess the intrinsic importance of each sentence based on its quality and informational content. This assessment establishes a priority order for expanding candidate steps into the search tree. Therefore, the role of BIS is to guide the global direction of the search.
> - **Local Direction (Cost Function):** At each step of the search, we examine only the top $W$ candidates from the BIS queue. We form a "thinking span" around each candidate and apply the cost function $f(\cdot)$ to select the single best span to advance the search path, returning the other candidates to the queue. This makes the Cost Function a local, step-by-step decision-maker.
>
> A key distinction lies in their nature: **`BIS scores are static`**, calculated once to reflect the intrinsic merit of each step. In contrast, the **`Cost Function is dynamic`**. Navigating a non-enumerable search space efficiently requires a dynamic function that can accurately assess a path's current cost and future value at every step.
>
> This conceptual separation informs our choice of models. For the global ranking task performed by BIS, we use the efficient and cost-effective GPT-2 model. For the critical, dynamic, and fine-grained local evaluation managed by the Cost Function, we leverage the more powerful and precise s1.1-32B model to achieve superior performance.
>
> # For Q 3.2
>
> The exploration size $W$ was set to 2 (already reported in the Training Details paragraph of the original Section 4.1, line 206).
>
> # For Q 3.3
>
> We use the Bidirectional Importance Score (BIS) to sort the sentences in **`descending order`**, forming a priority queue. In this queue, the sentences are intentionally out of sequence compared to their order in the original CoT.
>
> However, when they are selected and added to the compressed path, they are arranged to maintain their **`original order`**.
>
> # For Q 3.4
>
> After expanding the $W$ options, candidates that are not selected are **`put back into the queue`**.
>
> # For Q 4
>
> Given the current compressed thinking path  $\mathbf{t}^{\prime}_k$, question $\mathbf{q}$ and solution $\mathbf{s}$, the validation model $\mathcal{V}$ plays a dual role during the A* search phase: **generating a candidate solution $\mathbf{s}^{\prime}_k$ for verification and calculating the Cost Function $f(\cdot)$ for exploration** (reported in Section 3.1, lines 132-139 and 158-167).
>
> ## 1. Verification Phase
>
> First, the validation model $\mathcal{V}$, is prompted to perform sequence completion on the following input:
> $$\mathbf{q} \langle think \rangle \mathbf{t}_k^{\prime} \langle/think\rangle \langle solution \rangle$$
> This generates a candidate solution $\mathbf{s}^{\prime}_k$, resulting in the complete sequence:
> $$\mathbf{q} \langle think \rangle \mathbf{t}^{\prime}_k \langle /think \rangle \langle solution \rangle \mathbf{s}^{\prime}_k \langle /solution \rangle$$
> If the generated solution matches the ground truth ($\mathbf{s}^{\prime}_k==\mathbf{s}$), it indicates that the current path $\mathbf{t}^{\prime}_k$​ is of high quality and contains sufficient information. The algorithm then terminates and returns $\mathbf{t}^{\prime}_k$​ as the final output. If they do not match ($\mathbf{s}^{\prime}_k\neq\mathbf{s}$), the algorithm proceeds to the exploration phase.
>
> ## 2. Exploration Phase
>
> In this phase, the top $W$ "thinking spans" from the priority queue, denoted as $\lbrace \mathbf{r}_1, \dots, \mathbf{r}_W \rbrace$, are selected as candidates for expansion. For each candidate $\mathbf{r}_w$, the validation model $\mathcal{V}$ is used to evaluate the cost function $f\left (\mathbf{t}_k^{\prime}+\mathbf{r}_w\right) = g\left (\mathbf{t}_k^{\prime}+\mathbf{r}_w\right) + h\left (\mathbf{t}_k^{\prime}+\mathbf{r}_w\right)$. This is done by providing the model with the following input, which includes the ground-truth solution:
> $$\mathbf{q} \langle think \rangle \mathbf{t}^{\prime}_k+\mathbf{r}_w \langle /think \rangle \langle solution \rangle \mathbf{s} \langle /solution \rangle$$
> The candidate with the minimum cost function value is then chosen to expand the search tree, and the process moves to the next iteration. Further details on the algorithm and the validation model's operation are provided in Algorithm 1 in Appendix A, lines 382-383.
>
> # For Weakness
>
> To comprehensively evaluate its performance, in addition to the prompt-based baselines (Chain-of-Draft and Break-the-Chain in Appendix E), we also compared A*-Thought against a training-based baseline and the backbone model.
> - **Training-based baseline:** We compared against **`TokenSkip`**, a method that involves a two-stage process: first, shortening long CoT datasets with LLMLingua-2 compression, and second, using the resulting data to train an efficient model.
> - **Backbone model baseline:** We also included the QwQ-32B model as a fundamental point of comparison.

---

> > ### Comment · Reviewer_jq25 · 2025-08-01
> >
> > Thank you for your response and for addressing my confusion on many of the technical details.

---

### Official Review · Reviewer_hAiS · 2025-07-02

**Clarity:** 2
**Significance:** 3
**Originality:** 2
**Rating:** 5
**Confidence:** 4

**Summary:**

The paper proposes A*-Thought, an inference-time framework that prunes large language models’ long chain-of-thought (CoT) traces. Each step receives a bidirectional importance score (BIS) derived from attention statistics and token-level NLL with respect to both the question and the predicted answer. An A* search then selects a compact subset of spans that minimises a cost combining current-path verification loss and a heuristic future cost. Experiments on four math-reasoning benchmarks with three 32 B models show accuracy gains (up to +18 pp) and shorter outputs (≈50 % fewer tokens) under fixed token budgets compared with prompt- and training-based baselines.

**Questions:**

Q1: Have you tried GSM-like natural-language tasks, code reasoning, or commonsense QA to demonstrate domain robustness?

Q2: The verifier V is distilled from the same dataset used for pruning; could this inflate acceptance probability?

While the idea is interesting and results promising, the narrow evaluation scope, missing robustness analyses, and incomplete limitation discussion prevent a confident acceptance at this stage.

**Ethical Concerns:**

["NO or VERY MINOR ethics concerns only"]

**Final Justification:**

A*-Thought presents an interesting approach. The paper is well-written, demonstrating promising performance across datasets in different domains and achieving SOTA results.

**Limitations:**

They do not discuss: (a) sensitivity to α, β, k_max across tasks; (b) potential semantic drift when critical but low-BIS steps are pruned; (c) ethical risks of deploying compressed reasoning that may omit safety checks. Hence the limitations section is insufficient.

**Paper Formatting Concerns:**

N/A.

**Quality:**

2

**Strengths And Weaknesses:**

Strengths

- Potential to lower inference cost of LRMs, an important problem.

- Empirical improvements on math tasks under tight budgets (Table 1).

Weaknesses

- Evaluation limited to synthetic math datasets; no evidence on open-domain, multimodal or non-numeric reasoning.

- Variance/error bars omitted despite stochastic search; single-run results may be fragile.

- Conceptually close to Retro-Search and TokenSkip; contribution may be incremental (reuse of A* and attention/NLL saliency).

- Gains are shown only for already-compressed 32 B distillations; unclear impact on frontier ( >70 B) models or real latency on hardware.

- Training still demands multi-epoch passes over the entire CoT corpus (8 × A100 80 G, 3 epochs) — energy savings may be overstated.

---

> ### Author Rebuttal · Authors · 2025-07-30
>
> # For Question 1 / Weakness 1
>
> We thank the reviewer for their valuable suggestion. We tested A*-Thought's generalization on the out-domain benchmarks LiveCodeBench and MMLU across diverse domains such as **coding, history, computer science, law.**
>
> The results demonstrate that the model successfully learned the A*-Thought pattern, achieving higher efficiency and performance on **`non-mathematical tasks`**. We will incorporate these findings into the final version of the paper.
>
> |   Methods    |   MMLU   |            |  LiveCodeBench  |             | Average  |             |   ACU    |
> | :----------: | :------: | :--------: | :-------------: | :---------: | :------: | :---------: | :------: |
> |              |   Acc.   |    Len.    |      Acc.       |    Len.     |   Acc.   |    Len.     |          |
> |              |          |            | **Budget: 512** |             |          |             |          |
> |   QwQ-32B    |   37.6   |   511.94   |       0.0       |   512.00    |   18.8   |   511.97    |   3.67   |
> | + A*-Thought | **56.7** | **398.57** |     **4.5**     | **509.53**  | **30.6** | **454.05**  | **6.74** |
> |              |          |            | **Budget: 1K**  |             |          |             |          |
> |   QwQ-32B    |   57.4   |   956.90   |       0.0       |   1021.94   |   28.7   |   989.42    |   2.90   |
> | + A*-Thought | **71.9** | **573.30** |    **11.8**     | **986.02**  | **41.9** | **779.66**  | **5.37** |
> |              |          |            | **Budget: 2K**  |             |          |             |          |
> |   QwQ-32B    |   75.2   |  1323.26   |       3.5       |   1977.58   |   39.4   |   1650.42   |   2.38   |
> | + A*-Thought | **79.0** | **671.41** |    **24.5**     | **1734.28** | **51.8** | **1202.85** | **4.30** |
> |              |          |            | **Budget: 4K**  |             |          |             |          |
> |   QwQ-32B    |   79.5   |  1584.05   |      12.0       |   3586.93   |   45.8   |   2585.49   |   1.77   |
> | + A*-Thought |   **80.3**   |   **733.90**   |      **39.0**       |   **3044.51**   |   **59.7**   |   **1889.21**   |   **3.16**   |
>
> # For Question 2
>
> Verification model $\mathcal{V}$ is the original model of s1.1-32B, which was **fine-tuned on the original s1K-1.1 dataset** to preserve its long reasoning abilities. It is worth noting that the verification model was **`not fine-tuned on the compressed data`**.
>
> # For Limitation (a)
>
> We are grateful for the reviewer's insightful suggestion. Accordingly, we have performed a sensitivity analysis to explore the impact of varying the hyperparameters  $\alpha$, $\beta$, $k_{\mathrm{max}}$. This was done by evaluating the A*-Thought-QwQ-32B model on the ARC and LiveCodeBench benchmarks with 512 tokens budget. We will update the final version of our paper with these experimental results.
>
> |        Methods        | ARC  |        | LiveCodeBench |        | Average |        | ACU  |
> | :-------------------: | :--: | :----: | :-----------: | :----: | :-----: | :----: | :--: |
> |                       | Acc. |  Len.  |     Acc.      |  Len.  |  Acc.   |  Len.  |      |
> |     $\alpha=0.0$      | 65.9 | 405.31 |      5.5      | 509.31 |  35.70  | 457.31 | 7.81 |
> |     $\alpha=0.5$      | 63.5 | 381.02 |      4.5      | 509.53 |  34.00  | 445.28 | 7.64 |
> |     $\alpha=1.0$      | 65.0 | 393.20 |      8.5      | 505.59 |  36.75  | 449.40 | 8.18 |
> |      $\beta=0.1$      | 63.5 | 381.02 |      4.5      | 509.53 |  34.00  | 445.28 | 7.64 |
> |      $\beta=0.5$      | 52.5 | 438.03 |      4.0      | 510.73 |  28.25  | 474.38 | 5.96 |
> |      $\beta=0.9$      | 48.2 | 469.22 |      5.8      | 506.15 |  27.00  | 487.69 | 5.54 |
> | $k_{\mathrm{max}}=10$ | 35.3 | 476.90 |      2.2      | 506.68 |  18.75  | 491.79 | 3.81 |
> | $k_{\mathrm{max}}=15$ | 54.2 | 436.41 |      3.8      | 510.25 |  29.00  | 473.33 | 6.13 |
> | $k_{\mathrm{max}}=20$ | 63.5 | 381.02 |      4.5      | 509.53 |  34.00  | 445.28 | 7.64 |
>
> # For Limitation (b)
>
> Theoretically, Bidirectional Importance Score (BIS) is designed to find the most important reasoning steps. However, some key steps, like formulas, may get low scores when they're isolated.
>
> To address this, we propose the **`"thinking span" strategy`**. Instead of picking just one step $\mathbf{t}^{(n)}$ in A* search, we grab it along with its immediate neighbors:
> $$\mathbf{r}^{(n)}=\langle \mathbf{t}^{(n-1)}, \mathbf{t}^{(n)}, \mathbf{t}^{(n+1)}\rangle,$$
> where $\mathbf{t}^{(n-1)}$ and $\mathbf{t}^{(n+1)}$ are the immediately preceding and succeeding steps of $\mathbf{t}^{(n)}$. This method preserves local context, counteracting the effects of information fragmentation and **reducing potential semantic drift**.
>
> # For Limitation (c)
>
> Since the validation model is fine-tuned from the **`safety-aligned`** Qwen2.5-32B-Instruct, it also provides a degree of filtering for potentially harmful content during the A* search process.
>
> # For Weakness 2
>
> We thank the reviewer for their valuable suggestion. To make the experimental results more robust, we conducted **`multiple-runs and reasoned 5 times`**, **taking the average as the final result, and attaching the overall standard deviation after the result.**
>
> |   Methods    |   MATH500    |                  |    AMC23     |                  |  OlympiadBench  |                  |    GSM8K     |                 |   Average    |                  |     ACU      |
> | :----------: | :----------: | :--------------: | :----------: | :--------------: | :-------------: | :--------------: | :----------: | :-------------: | :----------: | :--------------: | :----------: |
> |              |     Acc.     |       Len.       |     Acc.     |       Len.       |      Acc.       |       Len.       |     Acc.     |      Len.       |     Acc.     |       Len.       |              |
> |              |              |                  |              |                  | **Budget: 512** |                  |              |                 |              |                  |              |
> |   QwQ-32B    | $10.8\pm0.7$ |  $512.00\pm0.0$  | $2.5\pm0.0$  |  $512.00\pm0.0$  |   $3.8\pm0.3$   |  $512.00\pm0.0$  | $28.7\pm1.4$ | $511.90\pm0.1$  | $11.5\pm0.5$ |  $511.98\pm0.0$  | $2.24\pm0.1$ |
> | + A*-Thought | $33.5\pm0.9$ |  $492.46\pm1.0$  | $18.5\pm2.0$ |  $509.00\pm2.9$  |  $11.4\pm0.6$   |  $509.00\pm0.7$  | $58.3\pm0.7$ | $453.84\pm2.0$  | $30.4\pm0.7$ |  $491.08\pm0.5$  | $6.19\pm0.1$ |
> |              |              |                  |              |                  | **Budget: 1K**  |                  |              |                 |              |                  |              |
> |   QwQ-32B    | $17.0\pm0.9$ | $1018.39\pm1.1$  | $11.0\pm4.9$ | $1024.00\pm0.0$  |   $6.4\pm0.2$   | $1023.97\pm0.0$  | $50.0\pm1.2$ | $950.39\pm2.4$  | $21.1\pm1.5$ | $1004.19\pm0.4$  | $2.10\pm0.2$ |
> | + A*-Thought | $50.3\pm2.1$ |  $853.22\pm4.3$  | $31.5\pm4.9$ | $944.83\pm14.7$  |  $20.3\pm1.1$   |  $962.95\pm6.3$  | $81.6\pm0.5$ | $696.85\pm4.6$  | $45.9\pm1.8$ |  $864.47\pm5.1$  | $5.31\pm0.2$ |
> |              |              |                  |              |                  | **Budget: 2K**  |                  |              |                 |              |                  |              |
> |   QwQ-32B    | $50.6\pm1.1$ | $1845.24\pm3.2$  | $26.0\pm1.2$ | $1970.87\pm9.5$  |  $18.0\pm1.0$   | $2020.94\pm2.3$  | $80.3\pm0.6$ | $1239.78\pm4.0$ | $43.7\pm0.5$ | $1769.21\pm3.9$  | $2.47\pm0.0$ |
> | + A*-Thought | $69.2\pm0.8$ | $1282.09\pm30.4$ | $44.0\pm5.6$ | $1509.43\pm40.0$ |  $31.1\pm1.1$   | $1640.21\pm8.0$  | $91.6\pm0.4$ | $835.68\pm5.6$  | $59.0\pm1.3$ | $1316.85\pm12.8$ | $4.48\pm0.1$ |
> |              |              |                  |              |                  | **Budget: 4K**  |                  |              |                 |              |                  |              |
> |   QwQ-32B    | $75.3\pm0.4$ | $2776.35\pm15.6$ | $50.0\pm6.3$ | $3485.04\pm39.6$ |  $35.8\pm0.8$   | $3634.92\pm9.4$  | $84.8\pm0.6$ | $1362.99\pm8.7$ | $61.5\pm1.6$ | $2814.83\pm11.8$ | $2.18\pm0.1$ |
> | + A*-Thought | $77.8\pm1.7$ | $1717.41\pm17.1$ | $60.0\pm4.5$ | $2286.07\pm49.9$ |  $41.4\pm1.9$   | $2535.87\pm28.0$ | $93.4\pm0.3$ | $883.02\pm17.6$ | $68.1\pm1.6$ | $1855.59\pm12.5$ | $3.67\pm0.1$ |
>
> # For Weakness 3
>
> To our knowledge, **`A*-Thought is the first work to adapt the principles of A* search for this task.`**
>
> Our method is fundamentally different from these predecessors.
>  - vs. Retro-Search: A*-Thought performs a **systematic, global search** based on importance. Retro-Search does a **local, sequential exploration**.
>  - vs. TokenSkip: Our BIS metric is **both question and solution aware**, whereas TokenSkip's is **agnostic to global context information**. We compress at the **sentence level** to preserve meaning, while TokenSkip works at the **token level**.
>  Thus, despite a shared objective, our methodology is unique.
>
> Beyond just guiding A* search, BIS metric also serves as a novel metric for analyzing the intrinsic importance mechanisms within long CoT. This aligns with a growing body of recent work (e.g., Token Entropy, Thought Anchors). We hope it provides a solid foundation for future research in this area.
>
> # For Weakness 4
>
> Following DeepSeek-R1-Distill-Qwen, s1, SkyThought, Bespoke-Stratos, 32B size is strong mid-scale model that is sufficient for long CoT. Based on this, we have centered our efforts on the 32B model scale.
>
> # For Weakness 5
>
> On our 8 A100 GPUs, training the s1-32B on its original 1K dataset took 3.84 hours, training Bespoke-Stratos-32B on its 17K dataset required 27 hours.
>
> In contrast, after using A*-Thought to compress the 1K-row dataset, reducing the average data length by **68.69%**. We trained our model for 3 epochs in **2.91 hours**. reducing the training time by **24.25%**, demonstrating a significant decrease in computational cost.

---

> > ### Comment · Reviewer_hAiS · 2025-08-01
> >
> > A*-Thought has demonstrated SOTA performance across multiple domains. After reviewing the comments from other reviewers, I find that all my concerns have been addressed. I believe this paper is acceptable and thus am happy to raise my score to 5 (Accept).

---

### Official Review · Reviewer_j7zW · 2025-07-03

**Clarity:** 3
**Significance:** 3
**Originality:** 3
**Rating:** 4
**Confidence:** 4

**Summary:**

This paper introduces A-Thought*, a framework that compresses lengthy Chain-of-Thought (CoT) outputs from large reasoning models using an A* search guided by a bidirectional importance score (BIS). The method addresses the efficiency challenges posed by verbose CoT reasoning, aiming to retain performance while significantly reducing token output.

**Questions:**

1. Can you explain why A*-Thought underperforms in high-budget scenarios? Is there an inherent trade-off in the compression strategy that degrades solution completeness?

2. How sensitive is the method to the selection of $\alpha$ (in BIS) and $\beta$ (in the cost function)? Have you performed any systematic hyperparameter tuning?

3. Have you tested A*-Thought on non-mathematical reasoning tasks (e.g., commonsense QA, multi-hop QA)? If not, what are the anticipated limitations?

**Ethical Concerns:**

["NO or VERY MINOR ethics concerns only"]

**Final Justification:**

Thanks for the authors' detailed response. I appreciate the results from the response and believe that it would make the paper more solid if the final version could include these results. However, I believe that my initial evaluation is accurate, as this paper is upper the borderline, therefore I decided to keep my original score.

**Limitations:**

Yes

**Quality:**

3

**Strengths And Weaknesses:**

## Strengths

1. **Clear Motivation and Relevance**: The problem of excessive reasoning length in LRMs is well-known, and addressing it aligns with growing interest in cost-efficient inference.

2. **Methodological Innovation**: The integration of a bidirectional importance score with A* search to identify high-value reasoning steps is novel.

3. **Empirical Performance**: Substantial gains are reported in accuracy and accuracy per computation unit (ACU), especially under constrained budgets (e.g., 2.39× improvement for QwQ-32B).

## Weaknesses

1. **Lack of Theoretical Analysis**: There is no rigorous theoretical justification for why BIS + A* would yield better CoT compression than alternatives.

2. **Limited Performance Gains in High-Budget Scenarios**: While A*-Thought shows impressive improvements under strict token constraints, its advantages diminish—and even reverse—when larger token budgets are available. In high-budget settings (e.g., 4096 tokens), A*-Thought underperforms compared to several baselines in terms of raw accuracy. This suggests that the proposed method may overly prioritize brevity at the cost of completeness.

---

> ### Author Rebuttal · Authors · 2025-07-30
>
> # For Question 1 / Weakness 2
>
> Based on the supplementary experiment ($k_{\mathrm{min}}=15$) and the original experiment ($k_{\mathrm{min}}=5$), it can be concluded that: **The trade-off is that a smaller $k_{\mathrm{min}}$ is more effective for low-budget reasoning, while a larger $k_{\mathrm{min}}$ is more effective for high-budget reasoning.**
>
> Due to the historical difficulties associated with achieving lossless compression at the text level, the trade-off between performance and efficiency is generally regarded as inevitable. The A*-Thought algorithm employs a minimum search depth,  $k_{\mathrm{min}}$ , to mitigate the negative semantic impact of overly short thought sequences. Verification of a thought path only occurs when its depth, $k$, meets the condition  $k \ge k_{\mathrm{min}}$.
>
> However, using an excessively low $k_{\mathrm{min}}$ value will result in the overall length of CoT data being too short, leading to performance degradation in high-budget scenarios. To address this, we explored increasing the value of $k_{\mathrm{min}}$​. The experiment results are shown in the following table. Notably, with $k_{\mathrm{min}}=15$ in **`4K budget`**, the A*-Thought model surpassed the performance of the strongest baseline, QwQ-32B fine-tuned with s1K-1.1, across all benchmarks. Furthermore, this best result was achieved with fewer inference tokens. We will update the final version of our paper with these findings.
>
> |              Methods               | MATH500  |             |   AMC23  |             | OlympiadBench |             |   GSM8K  |            |  Average |             |    ACU   |
> | :--------------------------------: | :------: | :---------: | :------: | :---------: | :-----------: | :---------: | :------: | :--------: | :------: | :---------: | :------: |
> |           **Budget: 4K**           |   Acc.   |     Len.    |   Acc.   |     Len.    |      Acc.     |     Len.    |   Acc.   |    Len.    |   Acc.   |     Len.    |          |
> |              QwQ-32B               |   75.4   |     2798.67 |     55.0 |     3456.05 |          36.5 |     3645.22 |     85.8 |    1348.24 |     63.2 |     2812.05 |     2.25 |
> |         QwQ-32B w/ s1K-1.1         |   79.6   |     2693.27 |     65.0 |     3485.95 |          42.4 |     3500.66 |     95.2 |    1624.11 |     70.6 |     2826.00 |     2.50 |
> | + A*-Thought $k_{\mathrm{min}}=5$  |   78.8   | **1699.78** |     65.0 | **2385.85** |          40.1 | **2546.45** |     93.1 | **874.54** |     69.3 | **1876.66** | **3.69** |
> | + A*-Thought $k_{\mathrm{min}}=15$ | **80.8** |     2184.34 | **67.5** |     2893.68 |      **44.2** |     3063.96 | **95.5** |    1229.40 | **72.0** |     2342.85 |     3.07 |
>
> # For Question 2
>
> For $\alpha$, which is used to balance between question and solution information within BIS, systematic hyperparameter tuning has been reported in Figure 9 of Appendix D, varying $\alpha$ across the discrete set of values **[0, 0.25, 0.5, 0.75, 1]**. Experiments show that the best model performance can be achieved when BIS effectively integrates problems and solutions.
>
> For $\beta$, it is used to adjust the weight of the current cost function $g(\cdot)$ in the overall cost function $f(\cdot)$. In the following supplementary experiments, we discussed its discrete values in **[0.1, 0.5, 0.9]** on the AI2 Reasoning Challenge (ARC) and LiveCodeBench. We will add these experimental results to the final version of the paper.
>
> |   Methods   | ARC  |        | LiveCodeBench |        | Average |        | ACU  |
> | :---------: | :--: | :----: | :-----------: | :----: | :-----: | :----: | :--: |
> |             | Acc. |  Len.  |     Acc.      |  Len.  |  Acc.   |  Len.  |      |
> | $\beta=0.1$ | 63.5 | 381.02 |      4.5      | 509.53 |  34.00  | 445.28 | 7.64 |
> | $\beta=0.5$ | 52.5 | 438.03 |      4.0      | 510.73 |  28.25  | 474.38 | 5.96 |
> | $\beta=0.9$ | 48.2 | 469.22 |      5.8      | 506.15 |  27.00  | 487.69 | 5.54 |
>
> # For Question 3
>
> We thank the reviewer for their valuable suggestion. To validate the generalization capabilities of the A*-Thought algorithm across diverse domains such as **coding, history, computer science, and law**, we conducted supplementary evaluations on the LiveCodeBench and MMLU benchmarks. It is important to note that our model was trained exclusively on mathematical tasks, making these benchmarks entirely out-domain.
>
> The results demonstrate that the model successfully learned the A*-Thought pattern, achieving higher inference efficiency and performance on **`non-mathematical tasks`**. We will incorporate these findings into the final version of the paper.
>
> |   Methods    |   MMLU   |            |  LiveCodeBench  |             | Average  |             |   ACU    |
> | :----------: | :------: | :--------: | :-------------: | :---------: | :------: | :---------: | :------: |
> |              |   Acc.   |    Len.    |      Acc.       |    Len.     |   Acc.   |    Len.     |          |
> |              |          |            | **Budget: 512** |             |          |             |          |
> |   QwQ-32B    |   37.6   |   511.94   |       0.0       |   512.00    |   18.8   |   511.97    |   3.67   |
> | + A*-Thought | **56.7** | **398.57** |     **4.5**     | **509.53**  | **30.6** | **454.05**  | **6.74** |
> |              |          |            | **Budget: 1K**  |             |          |             |          |
> |   QwQ-32B    |   57.4   |   956.90   |       0.0       |   1021.94   |   28.7   |   989.42    |   2.90   |
> | + A*-Thought | **71.9** | **573.30** |    **11.8**     | **986.02**  | **41.9** | **779.66**  | **5.37** |
> |              |          |            | **Budget: 2K**  |             |          |             |          |
> |   QwQ-32B    |   75.2   |  1323.26   |       3.5       |   1977.58   |   39.4   |   1650.42   |   2.38   |
> | + A*-Thought | **79.0** | **671.41** |    **24.5**     | **1734.28** | **51.8** | **1202.85** | **4.30** |
> |              |          |            | **Budget: 4K**  |             |          |             |          |
> |   QwQ-32B    |   79.5   |  1584.05   |      12.0       |   3586.93   |   45.8   |   2585.49   |   1.77   |
> | + A*-Thought |   **80.3**   |   **733.90**   |      **39.0**       |   **3044.51**   |   **59.7**   |   **1889.21**   |   **3.16**   |
>
> # For Weakness 1
>
> We provide a deeper theoretical analysis of A*-Thought. The rationality of "BIS + A* Search" can be explained by using **`importance sampling`**. The theoretical explanation is as follows:
>
> Given an LRM $\mathcal{M}$, it can generate an extended thinking trajectory comprising $N$ reasoning steps, denoted as $\mathbf{t} = { \mathbf{t}^{(1)}, \mathbf{t}^{(2)}, \dots, \mathbf{t}^{(N)}}$, and the corresponding solution $\mathbf{s}$ for a given question $\mathbf{q}$. This process can be represented as:
> $$( \mathbf{t}, \mathbf{s} ) := \mathcal{M}( \mathbf{q} ).$$
> We denote $\mathbf{s}$ as expectation $\mathbb{E}$ of $\mathcal{M}$ about $\mathbf{q}$ over the distribution $\mathbf{t}$:
> $$\mathbf{s} = \mathbb{E}_ {\mathbf{t}}^{\mathcal{M}}[\mathbf{q}] = \sum ( \mathbf{q} \mathbf{t} ).$$
> **Standard LRM using uniform sampling** of the thinking trajectories to approximate the expectation:
> $$\mathbb{E}_ {\mathbf{t}}^{\mathcal{M}} [ \mathbf{q} ] \approx \frac{1}{N} \sum_{i=1}^{N} \mathbf{q} \mathbf{t}^{(i)},$$
> if the final solution $\mathbf{s}$ is correct, we can call it an unbiased estimate. When $N$ is large enough, the variance of $\mathbf{s}$ sampled on $\mathbf{t}$ can be scaled by a factor of $N$ by the variance of $\mathbf{q}$ sampled uniformly on $\mathbf{t}$:
> $$\mathbb{V}_ {\mathbf{t}} [ \mathbf{s} ] = \frac{1}{N} \mathbb{V}_ {\mathbf{t}} [ \mathbf{q} ].$$
> However, as the number of thought steps $N$ keeps increasing, the model will frequently switch thought modes, leading to an increase in the variance of the thought trajectory. If such thought trajectories are sampled, it tends to reduce the convergence speed of the solution estimation and increase the chance of biased estimation.
>
> **BIS + A\* Search provides an approximate method for importance sampling**, which will introduce a new distribution $\mathbf{t}^{\prime}$, the new expectation can be derived from $\mathbb{E}_ {\mathbf{t}}^{\mathcal{M}} [ \mathbf{q} ]$:
> $$\mathbb{E}_ {\mathbf{t}}^{\mathcal{M}} [ \mathbf{q} ] = \sum  ( \mathbf{t}^{\prime} \frac{\mathbf{t}}{\mathbf{t}^{\prime}} \mathbf{q} ) = \mathbb{E}_ {\mathbf{t}^{\prime}}^{\mathcal{M}} [ \frac{\mathbf{t}}{\mathbf{t}^{\prime}} \mathbf{q} ],$$
> the expectation of $\mathcal{M}$ about $\mathbf{q}$ over the distribution $\mathbf{t}^{\prime}$ can be further transformed as:
> $$\mathbb{E}_ {\mathbf{t}^{\prime}}^{\mathcal{M}} [ \frac{\mathbf{t}}{\mathbf{t}^{\prime}} \mathbf{q} ] \approx \frac{1}{N} \sum_{i=1}^{N} \mathbf{q} \frac{\mathbf{t}^{(i)}}{\mathbf{t}^{\prime(i)}}.$$
> At this point an improvable variance about $\mathbf{s}$ is obtained:
> $$\mathbb{V}_ {\mathbf{t}^{\prime}} [ \mathbf{s} ] = \frac{1}{N} \mathbb{V}_ {\mathbf{t}^{\prime}} [ \frac{\mathbf{t}}{\mathbf{t}^{\prime}} \mathbf{q} ],$$
> making $\mathbb{V}_ {\mathbf{t}^{\prime}} [ \mathbf{s} ] < \mathbb{V}_ {\mathbf{t}} [ \mathbf{s} ]$. Given that $\mathbf{s}$ is sampled on $\mathbf{t}^{\prime}$, thus it can be achieved by minimizing the variance of $\mathbf{t}^{\prime}$:
> $$\mathbf{t}^{\prime} = \arg\min_{\mathbf{t}^{\prime}} \mathbb{V}_ {\mathbf{t}^{\prime}} [ \mathbf{s} ] = \arg\min_{\mathbf{t}^{\prime}} \frac{1}{N} \mathbb{V}_{\mathbf{t}^{\prime}} [ \frac{\mathbf{t}}{\mathbf{t}^{\prime}} \mathbf{q} ]$$
> Therefore, the ultimate goal is to find a proposal BIS distribution for the thought steps that minimizes this variance. By using the quality and information density of thought steps as heuristic information, the A* Search algorithm dynamically searches for the most important intermediate thought steps. This improves the quality of the final estimate while speeding up its convergence.

---

### Decision · Program_Chairs · 2025-09-17

**Decision:**

Accept (poster)

**Comment:**

This paper introduces A*-Thought, a framework that compresses lengthy Chain-of-Thought (CoT) outputs from large reasoning models using A* search guided by a bidirectional importance score (BIS). The method addresses efficiency challenges in verbose CoT reasoning by identifying high-value reasoning steps through attention statistics and token-level negative log-likelihood, aiming to retain performance while significantly reducing token output length.

The paper addresses a highly relevant problem of excessive reasoning length in large reasoning models, which aligns with growing interest in cost-efficient inference. The methodological innovation of integrating bidirectional importance scoring with A* search to identify essential reasoning steps is novel and well-motivated. Experimental results demonstrate substantial performance gains, with up to 2.39× improvement for QwQ-32B under constrained budgets and approximately 50% reduction in output tokens. The approach shows strong empirical performance across multiple mathematical reasoning benchmarks and demonstrates generalization capability across different large reasoning models. Thus I recommend to accept this paper.

Following the discussion phase, all major reviewer concerns have been adequately addressed. The authors successfully clarified the theoretical foundations, provided additional analysis of computational costs, and demonstrated broader applicability across different model scales. The novel combination of bidirectional importance scoring with A* search represents a valuable contribution to efficient reasoning in large language models, with strong empirical validation supporting its practical utility.